# Open-Set Recognition Interaction Effects: Modular Gains and Where to Find Them

## Abstract

Open-set recognition (OSR) requires neural networks to classify known classes while rejecting unknown samples, which is critical for real-world deployment. So far, OSR research studied and developed representation learning and postprocessing methods independently and their interaction effects remain unexplored, leaving potential performance gains untapped. In this paper, we present the first systematic study of these interactions across dataset scales and auxiliary data usage. First, we discover a failure mode we term *magnitude collapse*, where representation learning methods that utilize auxiliary data can suffer performance degradation at large scale and irreversibly destroy discriminative information, despite excelling at small scale. Second, we study the interaction effects between representation learning and postprocessing methods, and reveal when they can be leveraged for modular performance gains via two-stage processing. We also show where interaction effects amplify performance degradation due to magnitude collapse. Third, we show how these findings can be used to achieve state-of-the-art performance with a simple baseline and two-stage processing of OSR techniques. Finally, our results demonstrate that small-scale evaluations with auxiliary data are not predictive of large-scale performance, invalidating current best practices in OSR research.

## 1 Introduction

The rapid advancement of deep learning methods for image recognition increasingly promotes their real-world adoption, which requires them to adequately detect and handle unknown inputs for the reliability and safety of these systems (Scheirer et al., 2013; Hendrycks & Gimpel, 2017; Vaze et al., 2022). This task is typically studied under the two closely-related problem formulations: *Open-set Recognition* (OSR) and *Out-of-Distribution* (OOD) detection. Both aim to improve the robustness of classifiers by detecting distributional shifts in test-time samples.

While the categorization of OSR methods into Representation Learning (RL) and Post-Processing (PP) methods is commonly understood, current OSR methods are studied in isolation or compared as standalone methods, neglecting their modular nature and potential for improvements through combinations. Since the *interaction effects* between RL and PP have neither been explored nor formalized for OSR methods, we ask: can RL methods enhance or shape feature representations to amplify downstream PP performance? Or vice versa: is the optimal choice of PP method dependent on the RL training objective?

In both RL and PP methods, the feature magnitude has been identified as a crucial factor for performance (Dhamija et al., 2018; Hendrycks et al., 2022; Vaze et al., 2022; Cruz et al., 2024; Rabinowitz et al., 2025). For instance, Wang et al. (2025) highlight that *magnitude-aware* (MA) OOD postprocessing generally outperforms alternatives. This raises the question: do MA postprocessors synergize with RL methods that purposefully manipulate feature magnitudes during training? We identify a sub-category of RL methods, which we term *magnitude-manipulating* (MM), that utilize auxiliary data to separate known and auxiliary classes during training based on feature magnitude. However, MM methods tend to be sensitive to auxiliary data distribution at large scale, with performance falling below baselines,

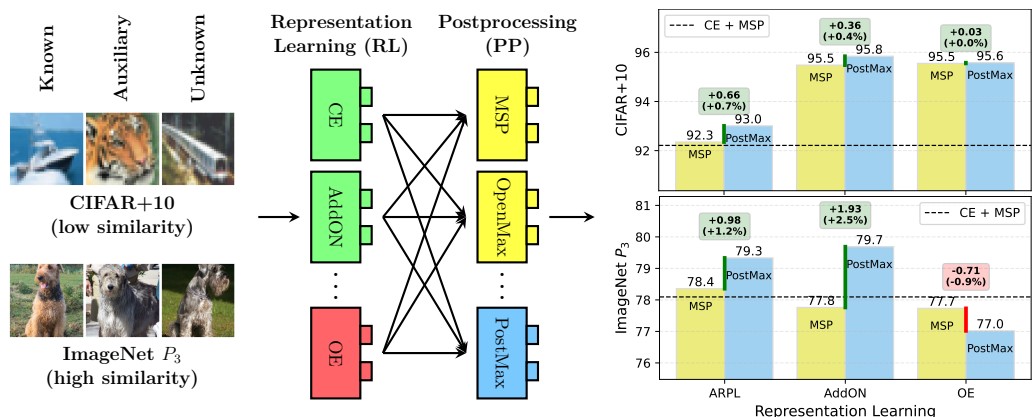

Figure 1: The modular two-stage OSR framework separates representation learning (RL) and postprocessing (PP), and reveals additional performance gains, by leveraging interaction effects between the two. Undesirable interaction effects exist for magnitude-manipulating RL (red) combined with magnitude-aware PP (blue), while beneficial effects are observed for AddON and magnitude-aware PP. Small-scale benchmarks are not predictive of large-scale performance and do not exhibit similar behavior due to limited similarity between known and auxiliary classes.

despite their success on small-scale datasets (Hendrycks et al., 2021; Wang et al., 2025). This raises concerns about their real-world applicability, and we seek to understand the underlying causes of this performance degradation at scale.

Our analysis reveals the mechanism behind the performance degradation of MM methods as the interplay of magnitude-manipulation and high similarity between auxiliary and known classes, a scenario not found on small-scale benchmarks. This causes a *magnitude collapse* in similar known classes and creates an undesirable linear dependency between feature magnitudes and class-wise accuracy, leading to systematically imbalanced class-wise detection performance. We show that magnitude collapse can be avoided by using an Additional Output Node (AddON) for auxiliary data, a simple and effective baseline that consistently outperforms other methods across scales and does not require hyperparameter tuning. This degradation of MM methods is further amplified by MA postprocessing, which otherwise experiences desirable interaction effects when combined with non-MM RL methods and outperforms other PP methods. Moreover, our experiments suggest that RL without auxiliary data and PP methods are highly independent components of OSR systems with clear separation of responsibilities, enabling modular additive performance gains.

Overall, our study advances the understanding of OSR systems and provides actionable guidelines for researchers and practitioners. First, leverage the modularity of OSR systems by augmenting non-MM RL methods with MA PP methods to achieve additional performance gains almost for free. Second, when training or fine-tuning a classifier with auxiliary data that is visually similar to known classes, avoid MM RL methods. Instead, use AddON as baseline to mitigate magnitude collapse and leverage positive interaction effects. Finally, validate OSR methods on large-scale benchmarks before deployment, as small-scale evaluations with auxiliary data are not predictive of large-scale performance.

In summary, our contributions are as follows:

- For the first time in OSR literature, we explore the modularity and interaction effects of representation learning and postprocessing methods, revealing where modular performance gains can be achieved and where to avoid negative synergies.

- We discover the magnitude collapse mechanism behind performance degradation at scale and how it impacts interaction effects.

- We demonstrate how interaction effects and auxiliary data *can* be leveraged at scale to achieve state-of-the-art performance regardless of auxiliary data distribution with the simple AddON baseline and two-stage processing of OSR techniques.
- Our experiments highlight that small-scale evaluations are not predictive of large-scale performance when using auxiliary data.

## 2 RELATED WORK

**OSR and Relation to OOD Detection.** OSR is formalized as the task of accurately classifying samples from known classes while rejecting samples from semantically unknown classes (Scheirer et al., 2013), thereby detecting test-time semantic shift (Vaze et al., 2022). OOD detection is a broader task (Yang et al., 2024) that aims to detect general distribution shift, which can include semantic or covariate shifts (Yang et al., 2024; Wang et al., 2025; Hendrycks et al., 2021) and is often posed as a binary classification problem (Hendrycks & Gimpel, 2017; Liang et al., 2017; Liu et al., 2020; Huang et al., 2021; Sun et al., 2021). As a result, OSR and OOD detection differ in their evaluation protocols (Vaze et al., 2022; Wang et al., 2025): OSR partitions a single dataset into known and unknown classes to remove covariate shifts (Neal et al., 2018; Palechor et al., 2023), while OOD detection typically uses different datasets for in-distribution (ID) and OOD classes (Hendrycks & Gimpel, 2017). Despite these differences, it has been indicated that methods that perform well on one task tend to perform well on the other (Vaze et al., 2022; Yang et al., 2024; Wang et al., 2025).

**Auxiliary Data in OSR.** Auxiliary samples serve as a proxy for unknown classes during training and are distinct from known or ID classes. Auxiliary samples are also referred to as known unknowns (Scheirer et al., 2014; Dhamija et al., 2018), outlier images (Kong & Ramanan, 2021), natural adversarial examples (Hendrycks et al., 2021), and negative samples (Palechor et al., 2023). Real auxiliary data is used for OOD detection (Hendrycks et al., 2019; Liu et al., 2020) and OSR (Dhamija et al., 2018; Palechor et al., 2023), dating back to the earliest approaches (Scheirer et al., 2014). While a large attention in OSR research is paid to artificially generate auxiliary samples (Ge et al., 2017; Neal et al., 2018; Chen et al., 2020; 2021), in this study we exclude generative methods and instead use real auxiliary data. The standard small-scale benchmarks MNIST, CIFAR, SVHN, and TinyImageNet partition all classes into known and unknown (Neal et al., 2018), therefore do not allow any auxiliary classes. The large-scale Semantic Shift Benchmark (SSB) (Vaze et al., 2022) uses the entire ImageNet-1K dataset as known classes and selects unknowns from a set of disjoint classes from ImageNet-21K-P.

**Differences and Similarities to Prior Art.** Wang et al. (2025) recently acknowledged the distinction between RL and PP methods and the potential for combined approaches in the context of disentangling OSR and OOD methods and benchmarks. They focus primarily on OOD detection methods, with 10 out of 12 methods being postprocessing, leaving modern OSR methods and their interaction effects unexplored.

## 3 MODULAR TWO-STAGE FRAMEWORK FOR OSR

In this paper, we disentangle OSR methods into modular sequential components: Representation Learning (RL) via classifier training and PostProcessing (PP) of pre-computed representations. Within this two-stage framework, every OSR system can be viewed as a combination of one RL and one PP method, denoted as RL+PP, *e. g.*, our baseline CE+MSP combines Cross-Entropy (CE) training with Maximum SoftMax Probability (MSP). We summarize key characteristics relevant to this study of RL and PP methods in Table 1. Section A.1 discusses how methods from Table 1 can be formalized in this framework.

**Representation Learning Methods.** RL methods train or fine-tune a classifier and extract the representations $\mathcal{R}_n = (\varphi_n, \mathbf{z}_n, \mathbf{y}_n)$ for sample $\mathbf{x}_n$, where $\varphi_n$ are discriminative deep features, $\mathbf{z}_n$ are the logits, and $\mathbf{y}_n$ the probability distributions over the known classes. RL methods can modify the training process in various ways, typically by adapting the loss

Table 1: Key characteristics of (a) Representation Learning (RL) and (b) Postprocessing (PP) methods for OSR. RL methods highlight which types of auxiliary data they use, whether they are Magnitude-Manipulating (MM), and whether they use an additional output node for the unknown class. For PP methods, we list whether they require training, are Magnitude-Aware (MA), and which types of inputs they operate on.

(a) Representation Learning

| Method | Auxiliary | MM | Output $K+1$ |
|---|---|---|---|
| Cross-Entopy (CE) | none | | |
| ARPL (Chen et al., 2021) | none | | |
| AddON (Palechor et al., 2023) | real | | Yes |
| Objectosphere (OS) (Dhamija et al., 2018) | real | Yes | |
| Outlier Exposure (OE) (Hendrycks et al., 2019) | real | Yes | |

(b) Postprocessing

| Method | Trainable | MA | Inputs |
|---|---|---|---|
| Maximum Softmax (MSP) (Hendrycks & Gimpel, 2017) | | | $\mathbf{y}$ |
| MaxLogits/MLS (Hendrycks et al., 2022; Vaze et al., 2022) | | Yes | $\mathbf{z}$ |
| OpenMax (Bendale & Boult, 2016) | Yes | | $\varphi$ |
| PostMax (Cruz et al., 2024) | Yes | Yes | $\varphi, \mathbf{z}$ |
| GHOST (Rabinowitz et al., 2025) | Yes | Yes | $\varphi, \mathbf{z}$ |

function (Dhamija et al., 2018; Hendrycks et al., 2019; Chen et al., 2020; 2021), involving data augmentation, such as mixup (Zhang et al., 2018; Verma et al., 2019) or generative methods (Ge et al., 2017; Neal et al., 2018; Verma et al., 2019; Kong & Ramanan, 2021; Chen et al., 2021; Wilson et al., 2023; Huang et al., 2023), or combinations thereof (Zhou et al., 2021). RL methods are trained on a dataset $\mathcal{K}_{\text{train}} \cup \mathcal{A}_{\text{train}}$, where for input $\mathbf{x}_n$, $\mathcal{K} = \{(\mathbf{x}_n, \tau_n) \mid \tau_n \in \mathcal{Y}\}$ is the set of known samples with known class labels $\mathcal{Y} = \{1, \ldots, K\}$, and $\mathcal{A} = \{(\mathbf{x}_n, \tau_n) \mid \tau_n \notin \mathcal{Y}\}$ is the set of auxiliary samples. Evaluation is done on $\mathcal{K}_{\text{test}} \cup \mathcal{U}_{\text{test}}$, where $\mathcal{U} = \{(\mathbf{x}_n, \tau_n) \mid \tau_n \notin \mathcal{Y}\}$ denotes unknown samples. Note that while auxiliary and unknown samples do not require a specific target label, they are required to not share the label space with known classes $\mathcal{Y}$ (Scheirer et al., 2013).

**Postprocessing Methods.** PP methods operate post-hoc on representations $\mathcal{R}_n$ to add open-set capabilities to a pre-trained closed-set classifier,[1] making them a cheap alternative to expensive RL training. However, PP methods cannot undo any damage to the feature representation learned by the pre-trained network, *e.g.*, when deep feature distributions $\varphi$ from known and unknown classes overlap, no PP method is able to separate those samples. Postprocessors can involve training a secondary classifier (Scheirer et al., 2014; Rudd et al., 2017), employing a statistics model (Bendale & Boult, 2016), modifying the inputs (Liang et al., 2017), or simply returning elements of $\mathcal{R}_n$ (Hendrycks & Gimpel, 2017; Hendrycks et al., 2022). We formalize postprocessors as follows: for test sample $\mathbf{x}_n^*$ with $\mathcal{R}_n^*$, we require a PP to produce two outputs $\mathcal{P}_n^* = (k_n^*, \gamma_n^*)$, where $k_n^* \in \mathcal{K}$ is the prediction of a known class label and $\gamma_n^*$ is an OOD score, where high $\gamma_n^*$ scores indicate known classes. In an operational setting, the OSR decision function can be defined as:

$$G_{\text{OSR}}(\mathcal{R}_n^*; \theta) = \begin{cases} k_n^* & \text{if } \gamma_n^* \geq \theta \\ \text{unknown} & \text{otherwise} \end{cases} \tag{1}$$

## 4 Study design and experimental setup

We choose five different RL approaches to cover a varied selection of models, based on whether the method requires auxiliary data and whether it (explicitly or implicitly) manip-

---

[1]Note that most PP methods perform class predictions $k^*$ based on the argmax of the logits (or monotonic transformations thereof) and therefore yield identical class predictions and closed-set accuracy, addressing exclusively the separation between known and unknown classes.

ulates the feature magnitude. As such we selected the following methods: Cross-Entropy (CE), ARPL (Chen et al., 2021), AddON (Palechor et al., 2023), Outlier Exposure (OE) (Hendrycks et al., 2019), and ObjectoSphere (OS) (Dhamija et al., 2018). We also choose five different PP methods to cover a varied selection of methods based on whether it takes the feature magnitude into account, *i. e.*, it is magnitude-aware. We select MSP (Hendrycks & Gimpel, 2017), MaxLogits/MLS (Hendrycks et al., 2022; Vaze et al., 2022), OpenMax (Bendale & Boult, 2016), PostMax (Cruz et al., 2024) and GHOST (Rabinowitz et al., 2025).

**Datasets** We conduct small-scale experiments on the standard OSR **CIFAR+N** benchmarks with $N \in \{10, 50\}$ (Neal et al., 2018). These protocols randomly sample 4 known classes from CIFAR-10 and N unknown classes from CIFAR-100 Krizhevsky & Hinton (2009).[2] To allow training with auxiliary samples, we randomly sample N auxiliary classes from the remaining classes in CIFAR-100. Large-scale experiments are conducted on the **ImageNet** protocols $P_1$, $P_2$, and $P_3$ (Palechor et al., 2023) based on the ILSVRC 2012 dataset (Russakovsky et al., 2015). These protocols offer increasing levels of difficulty from $P_1$ to $P_3$ by increasing semantic similarity between known, auxiliary, and unknown classes based on the WordNet hierarchy (Miller, 1998). While $P_1$ poses an easy open-set task with low similarity between known and auxiliary classes, $P_2$ and $P_3$ pose increasingly difficult open-set tasks with high similarity between known and auxiliary classes.

**Evaluation Metrics** To evaluate the binary unknown rejection and the closed-set performance in isolation, we use the Area Under the Receiver Operating Characteristics (AUROC) curve and closed-set accuracy, respectively. To evaluate OSR performance we use Correct Classification Rate (CCR) (Dhamija et al., 2018) and False Positive Rate (FPR) and their single-valued derivations: Area Under the Open-Set Classification Rate (AUOSCR) curve (Vaze et al., 2022), which provides a threshold-agnostic measure, and the Operational Open-set Accuracy (OOSA) (Cruz et al., 2024), which equally weights known and unknown samples and measures performance at an operational threshold. For detailed descriptions of all metrics, please refer to Section A.5 in the appendix. Given knowns $\mathcal{K}$, unknowns $\mathcal{U}$, predictions $\mathcal{P}_n^* = (k_n^*, \gamma_n^*)$, and threshold $\theta$ we compute CCR and FPR following the adjustments by (Rabinowitz et al., 2025) to allow for arbitrary OOD scores $\gamma^*$:

$$
\begin{aligned}
\mathrm{CCR}(\theta) &= \frac{\left|\{(\mathbf{x}_n, \tau_n) \in \mathcal{K} \wedge k_n^* = \tau_n \wedge \gamma_n^* \geq \theta\}\right|}{|\,\mathcal{K}\,|} \\
\mathrm{FPR}(\theta) &= \frac{\left|\{(\mathbf{x}_n, \tau_n) \in \mathcal{U} \wedge \gamma_n^* \geq \theta\}\right|}{|\,\mathcal{U}\,|}
\end{aligned}
\tag{2}
$$

To evaluate our methods we use multiple metrics to capture different aspects of performance. AUOSCR and OOSA lack the interpretability of closed-set accuracy and AUROC, and therefore should not be solely relied upon. On the other hand, the latter do not measure OSR performance holistically (Wang et al., 2022). For example, closed-set accuracy is incapable of measuring interaction effects because all PP methods except OpenMax exclusively rely on the logit order from upstream RL methods to produce predictions $k^*$ and ignore $\gamma^*$ resulting in identical accuracy scores (Figure 11d in the appendix). AUROC measures the ability to separate known and unknown samples across all thresholds via $\gamma^*$ and ignores $k^*$. AUOSCR and OOSA however jointly measure the quality of $k^*$ and $\gamma^*$ for any given threshold $\theta$ via $\mathrm{CCR}(\theta)$ and $\mathrm{FPR}(\theta)$. As such we rely primarily on AUOSCR and OOSA to answer our research questions but draw complementary insights from closed-set accuracy and AUROC.

**Training Details** We train all networks from scratch to ensure that no information from unknown classes is leaked into training of pre-trained networks, and to isolate the effect of representation learning as opposed to fine-tuning a closed-set network. All networks are trained using SGD with momentum of 0.9 and an initial learning rate of 0.1 with cosine annealing schedule (Loshchilov & Hutter, 2017). For large-scale experiments we train

---

[2]We use the same class allocations as Chen et al. (2021) for comparability.

**AUOSCR**

CIFAR+10

| | MSP | OpenMax | MaxLogits | PostMax | GHOST |
|---|---|---|---|---|---|
| CE | 92.2 | 92.9 | 94.4 | 92.8 | 93.0 |
| ARPL | 92.3 | 93.0 | 94.5 | 93.0 | 93.0 |
| AddON | 95.5 | 95.4 | 96.2 | 95.8 | 96.0 |
| OE | 95.5 | 95.7 | 95.6 | 95.6 | 95.6 |
| OS | 95.8 | 95.8 | 95.9 | 95.4 | 95.8 |

CIFAR+50

| | MSP | OpenMax | MaxLogits | PostMax | GHOST |
|---|---|---|---|---|---|
| CE | 90.4 | 91.2 | 92.5 | 90.7 | 90.0 |
| ARPL | 90.6 | 91.3 | 92.7 | 91.0 | 90.2 |
| AddON | 96.3 | 95.2 | 95.9 | 96.2 | 96.0 |
| OE | 96.2 | 96.4 | 96.3 | 96.2 | 96.2 |
| OS | 96.2 | 96.2 | 96.2 | 95.6 | 96.2 |

ImageNet $P_1$

| | MSP | OpenMax | MaxLogits | PostMax | GHOST |
|---|---|---|---|---|---|
| CE | 75.6 | 76.5 | 76.8 | 76.7 | 77.1 |
| ARPL | 76.4 | 76.8 | 77.7 | 77.5 | 77.9 |
| AddON | 77.5 | 77.7 | 78.1 | 77.7 | 78.2 |
| OE | 77.4 | 77.3 | 77.8 | 77.4 | 77.8 |
| OS | 77.7 | 77.4 | 78.0 | 77.9 | 78.0 |

ImageNet $P_2$

| | MSP | OpenMax | MaxLogits | PostMax | GHOST |
|---|---|---|---|---|---|
| CE | 64.9 | 66.3 | 68.5 | 68.8 | 69.1 |
| ARPL | 66.1 | 66.7 | 69.2 | 69.8 | 70.1 |
| AddON | 68.5 | 70.5 | 71.5 | 71.8 | 72.2 |
| OE | 66.9 | 67.1 | 66.9 | 67.9 | 67.6 |
| OS | 67.8 | 67.9 | 68.0 | 68.4 | 68.1 |

ImageNet $P_3$

| | MSP | OpenMax | MaxLogits | PostMax | GHOST |
|---|---|---|---|---|---|
| CE | 78.1 | 77.9 | 77.9 | 78.9 | 78.8 |
| ARPL | 78.4 | 78.3 | 78.4 | 79.3 | 79.2 |
| AddON | 77.8 | 78.5 | 78.7 | 79.7 | 79.6 |
| OE | 77.7 | 77.8 | 77.1 | 77.0 | 77.8 |
| OS | 78.1 | 78.0 | 77.6 | 76.9 | 78.3 |

**OOSA**

CIFAR+10

| | MSP | OpenMax | MaxLogits | PostMax | GHOST |
|---|---|---|---|---|---|
| CE | 87.8 | 88.4 | 90.5 | 88.3 | 90.0 |
| ARPL | 88.0 | 88.6 | 90.7 | 88.6 | 90.2 |
| AddON | 92.2 | 91.9 | 93.7 | 93.0 | 93.6 |
| OE | 92.5 | 92.7 | 92.6 | 92.8 | 92.6 |
| OS | 92.7 | 92.7 | 92.7 | 92.5 | 92.6 |

CIFAR+50

| | MSP | OpenMax | MaxLogits | PostMax | GHOST |
|---|---|---|---|---|---|
| CE | 84.4 | 85.5 | 86.0 | 84.0 | 85.5 |
| ARPL | 85.0 | 85.2 | 86.4 | 84.9 | 86.0 |
| AddON | 93.6 | 91.8 | 92.3 | 93.1 | 92.8 |
| OE | 93.4 | 93.6 | 93.6 | 93.4 | 93.6 |
| OS | 93.9 | 93.8 | 93.9 | 92.5 | 94.0 |

ImageNet $P_1$

| | MSP | OpenMax | MaxLogits | PostMax | GHOST |
|---|---|---|---|---|---|
| CE | 81.0 | 83.5 | 83.8 | 84.2 | 85.0 |
| ARPL | 80.7 | 83.4 | 84.4 | 84.0 | 85.3 |
| AddON | 78.8 | 85.6 | 86.8 | 85.2 | 87.1 |
| OE | 83.3 | 82.8 | 83.3 | 83.2 | 85.1 |
| OS | 82.6 | 83.3 | 85.1 | 85.8 | 83.0 |

ImageNet $P_2$

| | MSP | OpenMax | MaxLogits | PostMax | GHOST |
|---|---|---|---|---|---|
| CE | 69.5 | 70.3 | 72.1 | 73.0 | 73.9 |
| ARPL | 69.8 | 70.5 | 72.1 | 74.5 | 74.2 |
| AddON | 66.2 | 74.6 | 76.1 | 76.3 | 76.4 |
| OE | 70.0 | 70.5 | 69.5 | 71.4 | 70.1 |
| OS | 70.1 | 70.1 | 70.1 | 72.1 | 69.5 |

ImageNet $P_3$

| | MSP | OpenMax | MaxLogits | PostMax | GHOST |
|---|---|---|---|---|---|
| CE | 77.9 | 77.7 | 77.9 | 78.8 | 78.3 |
| ARPL | 77.7 | 77.7 | 77.7 | 78.6 | 78.3 |
| AddON | 71.0 | 77.1 | 77.5 | 79.3 | 78.5 |
| OE | 72.9 | 73.1 | 73.0 | 75.7 | 74.0 |
| OS | 75.3 | 75.5 | 74.8 | 71.8 | 75.0 |

Figure 2: OSR performance for RL+PP combinations across datasets in AUOSCR (top) and OOSA (bottom). The color of each heatmap is normalized independently and centered at the CE+MSP baseline, where blue shows an increase and red a decrease. Baseline RL and PP methods are surrounded by a black border. Results for CIFAR+N are averaged over 5 trials.

ResNet50 for 120 epochs with batch size of 32 and weight decay of 1$e$-4. The small-scale experiments are trained with the CNN architecture from Neal et al. (2018); Chen et al. (2021) for 100 epochs with batch size of 128 and weight decay of 5$e$-4. We perform early stopping according to validation confidence (Palechor et al., 2023). For CE and ARPL, we compute validation confidence on known classes only since including them yielded unreliable results. Where possible, we rely on recommended hyperparameters for each RL and PP method. Others are optimized on the validation set via grid search. Optimal hyperparameters and their ranges are reported in Section A.4 in the appendix. All experiments are run on NVIDIA RTX GeForce 2080 Ti.

## 5 RESULTS AND DISCUSSION

### 5.1 OSR REPRESENTATION LEARNING WITH AUXILIARY DATA AT LARGE SCALE

The first set of experiments aims at answering if RL with auxiliary data can improve OSR performance on large-scale datasets, despite recent studies that suggest otherwise (Wang et al., 2025). We compare the performance of RL methods with auxiliary data (AddON, OE, and OS) to methods that only utilize known classes (CE and ARPL) across datasets. Here, we ignore interaction effects and consider MSP postprocessing or aggregate results over all postprocessors. The AUOSCR and OOSA for every RL+PP combination are shown in Figure 2, other metrics are reported in Tables 3 and 4 in the Appendix.

**Small-scale Outperformace with Auxiliary Data.** On CIFAR+N, all RL methods utilizing auxiliary data dramatically outperform those that do not by up to 5.9 percentage points in AUOSCR with MSP. State-of-the-art ARPL achieves consistent but negligible improvements over CE for any given PP. With known classes being held constant between CIFAR+10 and CIFAR+50, we can see that additional auxiliary data consistently improves AUOSCR by over 2 percentage points, even when evaluated on more unknown classes. For all RL methods, the variations across PP are comparably small, suggesting that RL contributes more toward overall performance than PP on small-scale benchmarks. Only MaxLogits provides substantial gains, up to 2.2 percentage points of AUOSCR over MSP for CE.

**Performance Degradation at Large Scale.** On large-scale ImageNet protocols, this outperformance from using auxiliary data vanishes, even underperforming the CE+MSP baseline on $P_3$ for most postprocessors, supporting Wang et al. (2025). We isolate the effect of RL by computing the *RL contribution delta* to the CE baseline as a function of

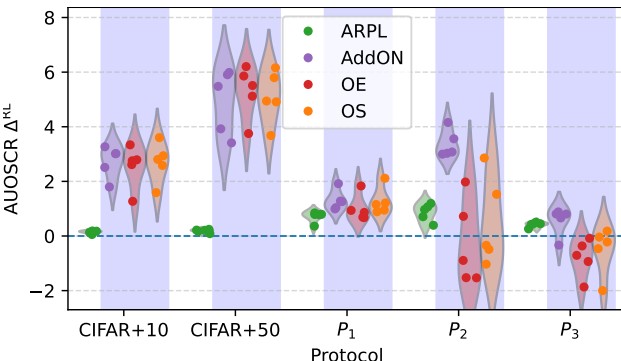

Figure 3: The RL performance contribution $\Delta^{\mathrm{RL}}$ in AUOSCR is shown as distributions over PP methods and across protocols, with $P_1$, $P_2$, and $P_3$ increasing in similarity between known and auxiliary classes. Methods that use auxiliary data are marked with blue background. CIFAR+N results are averaged over 5 trials.

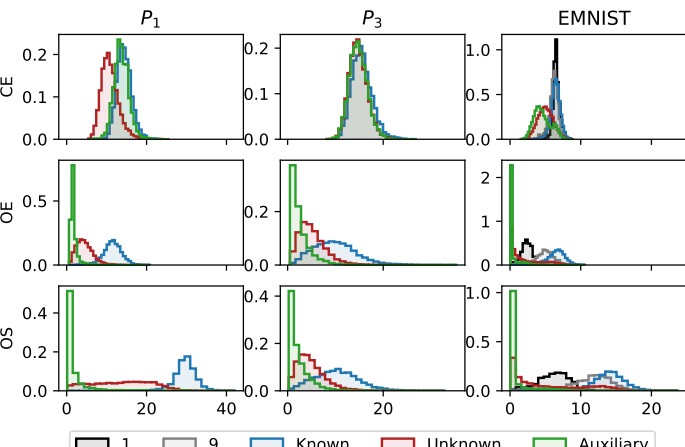

Figure 4: Feature magnitude distributions of known, auxiliary, and unknown classes on protocols $P_1$, $P_3$, and EMNIST. OE and OS experience feature magnitude collapse on $P_3$ and EMNIST, pulling their feature magnitudes towards zero. For EMNIST, we show distributions for known classes 1 and 9 (black and grey) that are highly similar to auxiliary classes, and other known classes (blue) separately.

the postprocessor PP. Similarly, we separate improvements of PP by computing the *PP contribution delta* to the MSP baseline:

$$\Delta^{\mathrm{RL}}_{\mathrm{method}}(\mathrm{PP}) = \text{"method+PP"} - \text{"CE+PP"}$$
$$\Delta^{\mathrm{PP}}_{\mathrm{method}}(\mathrm{RL}) = \text{"CE+method"} - \text{"RL+MSP"}$$

(3)

This allows us to decompose the gains from any OSR system RL+PP to the CE+MSP baseline, *e.g.*, on $P_1$ we have "ARPL+GHOST" − "CE+MSP" = $\Delta^{\mathrm{RL}}_{\mathrm{ARPL}}(MSP)$ + $\Delta^{\mathrm{PP}}_{\mathrm{GHOST}}(ARPL) \approx 0.8 + 1.5 = 2.3$. Figure 3 depicts the RL contribution deltas across datasets as distribution over all postprocessors. The RL contribution delta for MM methods OE and OS degrades and turns negative with increasing similarity of known and auxiliary classes on $P_2$ and $P_3$, destroying performance across most PP. Wang et al. (2025) attribute their findings of poor OE performance to low correlation between auxiliary and unknown classes or high correlation between known and unknown classes. However, AddON does not experience this performance degradation with identical data, demonstrating that it cannot be explained by the data distributions alone but by the interplay between data distributions and training objective. Strong OOD detection performance with MSP (see AUROC in Figure 11c) and high correlation between AUOSCR and closed-set accuracy (cf. Figure 11d in Appendix) suggest that the performance degradation can partially be explained by a loss of discriminative information for known classes.

**The Risk of Magnitude-Manipulation: Magnitude Collapse.** To understand *why* MM methods degrade in accuracy, AUOSCR, and OOSA, we analyze the feature magnitude distributions of known, auxiliary, and unknown samples (Figure 4). With highly similar auxiliary samples, MM methods inadvertently draw features of known classes towards the origin of the feature space, resulting in *magnitude collapse* and effectively overlapping them with auxiliary, unknown, and other known samples. We analyze the relationship between

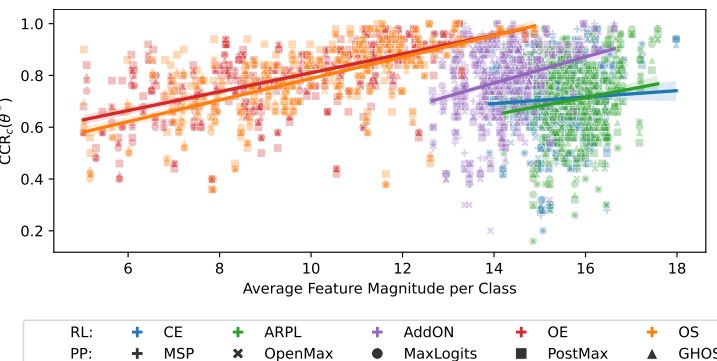

Figure 5: Linear relationship between class-wise CCR, $CCR_c(\theta^\star)$, at the operational threshold and class-wise average feature magnitude for known classes on $P_3$. We perform regression for each RL method independently and over all PP methods.

feature magnitude and class-wise CCR (Rabinowitz et al., 2025) at the operational threshold, $CCR_c(\theta^\star)$, see Section A.5, and perform linear regression[3] against the class-wise average feature magnitude for each known class $c$ (Figure 13 in the appendix). On CIFAR+N, only MM methods exhibit a statistically significant positive correlation between average feature magnitude and $CCR_c(\theta^\star)$, with large effect sizes of $R^2$ up to 80%. On the easily separable $P_1$, most models exhibit significant positive correlations, but often with meaningless effect sizes below 10%. However, with increasing similarity in known and auxiliary samples on $P_2$ and $P_3$, MM methods learn much stronger and statistically significant relationships, resulting in practically significant effect sizes of $R^2$ up to 38% on $P_3$ (Figure 5). This shows how magnitude-manipulation increases the dependency between feature magnitude and CCR by systematically reducing performance on a few classes in a trade-off to maintain *overall* high binary ID-vs-OOD separation via MSP (see AUROC Figure 11c) and increasing the minimal class-wise CCR.

In contrast, AddON counteracts magnitude collapse by learning sufficiently large feature magnitudes to achieve high SoftMax probabilities for the additional output node during training. Since AddON is trained via CE loss, it forces probabilities corresponding to any output node (known or auxiliary) to be close to 1 for samples of the respective class. Formally, we want the trained network to be able to achieve $\mathbf{y}_c > 1 - \epsilon$ for any sample and output class $c \in \{1, \ldots, C\}$, for some small $\epsilon > 0$. This holds if (but not only if) the difference between the maximal two logits surpasses a lower bound $l$, which is equivalent to (see Section A.2):

$$\|\varphi\|_2 > \frac{l}{\delta_{\mathbf{z}_c/\|\varphi\|_2}} = \frac{\log(\frac{1-\epsilon}{\epsilon}) + \log(C-1)}{\|\mathbf{W}_c\|_2 \cos(\alpha_c) - \max_{c' \neq c}\{\|\mathbf{W}_{c'}\|_2 \cos(\alpha_{c'})\}} \implies \mathbf{y}_c > 1 - \epsilon, \quad (4)$$

where $\delta_{\mathbf{z}_c/\|\varphi\|_2}$ is the difference logits divided by their feature magnitude. Note that $l$ is constant for any dataset, loss and $\epsilon$, and $\delta_{\mathbf{z}_c/\|\varphi\|_2}$ is generally small because weight magnitudes are minimized through weight decay and the cosines are bounded between -1 and 1. This provides upward pressure on feature magnitudes during and requires deep feature magnitudes to exceed a lower bound on the feature magnitude for any sample given a trained classifier. In other words, learning sufficiently large feature magnitudes is a sufficient condition to achieve high confidence for some class given trained classifier weights $\mathbf{W}$. This training incentive prevents magnitude collapse in AddON and ensures that AddON learns feature magnitude distributions similar to CE and ARPL. Surprisingly, despite training on auxiliary data, AddON learns lower feature magnitudes for auxiliary and unknown classes while maintaining sufficiently large feature magnitudes to achieve high confidences for known and auxiliary classes. The difference in feature magnitudes between known, auxiliary, and unknown classes is analogous to CE but slightly more pronounced, esp. on $P_3$. (Figure 10). Furthermore, this provides theoretical support for combinations of AddON with MA PP methods, because assumptions as well as high level magnitude distribution statistics are similar to CE, for which those PP methods were designed.

**Qualitative Analysis on Class Similarity.** We replicate and qualitatively investigate the feature collapse for MM methods at small scale by curating a hard EMNIST benchmark

---

[3]All regression assumptions are reasonably met.

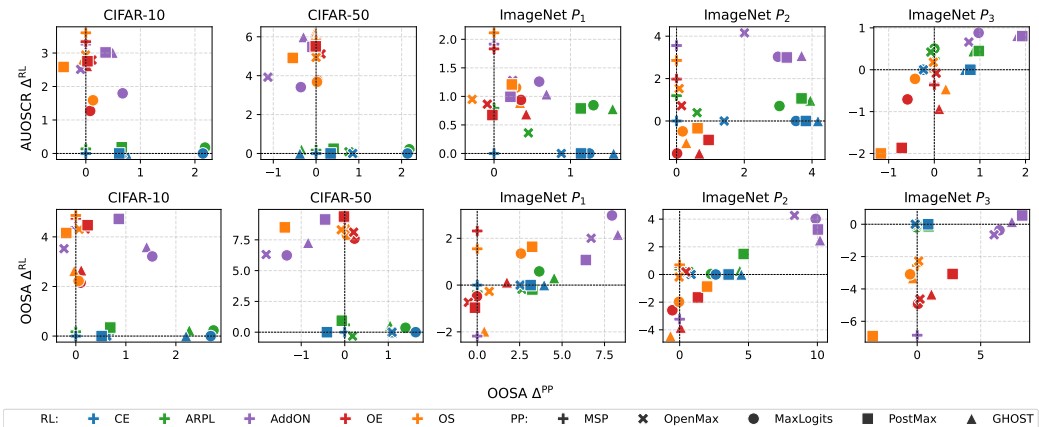

Figure 6: Interaction effects of RL and PP components as correlation between RL contribution $\Delta^{\mathrm{RL}}$ and PP contribution $\Delta^{\mathrm{PP}}$ for AUOSCR (top row) and OOSA (bottom row).

that includes auxiliary classes which are highly similar to known classes 1 and to a lesser extent 9 or even indistinguishable without context (see Section A.3 for details). Experiments are identical to CIFAR+N but networks are trained for 50 epochs. Observable from Figure 4 (right), for MM methods, the feature representations of digits 1 and 9 are pulled towards the origin. We compute $\mathrm{CCR}_c(\theta^\star)$ per RL method for classes 1, 9, and rest (all other known classes). For OS and OE combined, these classes experience $\mathrm{CCR}_c(\theta^\star)$ in the range [58.1%, 88.7%] (digit 1) and [93.6%, 97.2%] (digit 9), while other digits achieve over 96.7% CCR, showing dramatically imbalanced detection performance on known classes. See Table 5 in the appendix for all RL methods. Non-MM methods have $\mathrm{CCR}_c(\theta^\star)$ evenly distributed between 83.8% and 98% over all postprocessors. This clearly demonstrates how high similarity of known and auxiliary classes in conjunction with magnitude-manipulation irreparably damages the feature representation, and ruin the consecutive classification of known classes with similar appearance.

## 5.2 INTERACTION EFFECTS

The second set of experiments aims to answer whether the optimal PP method should be informed by the RL method, and whether magnitude-manipulating RL can enhance the performance of magnitude-aware PP methods. We analyze the relationship and interactions between RL and PP contributions by plotting the RL contribution delta $\Delta^{\mathrm{RL}}$ against the PP contribution delta $\Delta^{\mathrm{PP}}$ for each RL+PP combination in Figure 6.

**Independent Contributions for RL without Auxiliary Data.** Across all evaluation metrics and protocols, all experiments reveal that PP contributions are almost perfectly independent from RL contributions, when trained *without* auxiliary data, *e. g.*, the PP contribution of GHOST is independent of the used RL method: $\Delta^{\mathrm{PP}}_{\mathrm{GHOST}}(CE) \approx \Delta^{\mathrm{PP}}_{\mathrm{GHOST}}(ARPL) = 1.5$. This suggests that OSR system components have separate responsibilities when trained on known classes only, with RL addressing the ID classification and PP addressing the open-set capabilities.[1] This allows to combine any RL with any PP method, with magnitude-aware PP consistently favored (Wang et al., 2025), to achieve additive performance gains without the risk of undesirable interactions.

**Interaction Effects for RL with Auxiliary Data.** RL methods that train *with* auxiliary data only show interaction effects with high similarity of known and auxiliary classes. Interaction effects are characterized by a correlation between $\Delta^{\mathrm{RL}}$ and $\Delta^{\mathrm{PP}}$ in Figure 6. On ImageNet benchmarks, RL methods with auxiliary data show interaction effects, with particularly MA PP methods strongly amplifying positive and negative gains from RL. While magnitude-aware PP consistently outperform others for non-MM methods, they can amplify performance degradation for MM methods due to their sensitivity to the magnitude collapse.

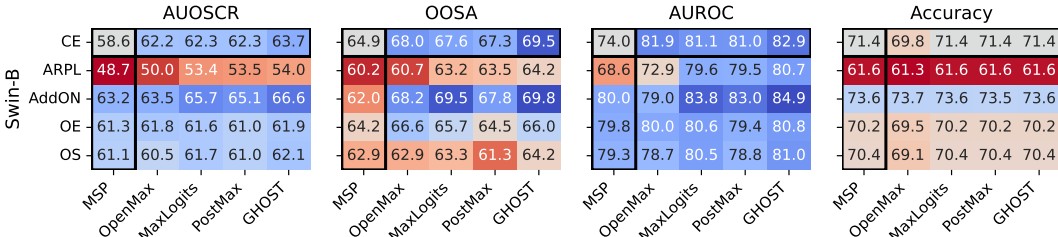

Figure 7: OSR performance on $P_2$ with Swin-B backbone, showing accuracy, AUROC, and AUOSCR metrics. Similar interaction effects between RL and PP methods are observed as with ResNet-50 backbones. Note that ARPL training was unstable and did not converge properly.

In particular, PostMax, and to a lesser extent GHOST, demonstrate both the highest gains for non-MM methods (8.3 percentage points or +11.7% for AddON), as well as the most severe performance degradation for MM methods (-1.2 percentage points or -1.5% for OS) on $P_3$. These interaction effects are even clearer when evaluated via OOSA, which equally weights known and unknown test samples.

Generally speaking, MM methods across all protocols did not benefit significantly from PP methods. Furthermore, we can see a clear trend, where small-scale experiments are primarily driven by RL via inclusion of auxiliary data and favor the simpler PP methods like MaxLogits, while large-scale experiments benefit from combining RL and PP, mainly through joining AddON with MA PP methods like PostMax or GHOST.

**Robustness to Backbone Architecture.** To ensure that our findings are not specific to the chosen ResNet-50 architecture, we replicate the experiments on $P_2$, which balances difficulty and computational cost, using a Swin-B transformer (Liu et al., 2021). Training is conducted with original hyperparameters, except for a reduced learning rate of 0.0002 to account for a lower batch size of 90. While overall performance is lower compared to ResNet-50 backbones, likely due to the increased data requirement of a transformer, we observe similar interaction effects between RL and PP methods as with ResNet-50 (Figure 7), where accuracy and AUROC are slightly lower for MM methods while AddON performed best with MA PP methods across all metrics. Notably, ARPL training was unstable and did not converge with an a priori selected seed and identical training parameters, resulting in significantly lower performance.

## 6 CONCLUSION

In this study, we adopt a two-stage framework for systematically disentangling Representation Learning (RL) and PostProcessing (PP) methods for Open-Set Recognition (OSR). We show that RL *without auxiliary data* leads to independent OSR components, that can be freely combined to achieve additive performance gains, whereas RL *with auxiliary data* can experience interaction effects with PP methods that can improve or degrade OSR performance. We explain this performance degradation and the key role of feature magnitude in interaction effects via the magnitude collapse mechanism, revealing several insights. First, the similarity between auxiliary and known classes is a key factor for performance degradation at scale, a scenario that does not occur on small-scale benchmarks. Second, magnitude collapse creates an undesirable linear dependency between feature magnitude and class-wise detection performance on the in-distribution and OSR task, leading to systematically imbalanced detection across known classes. However, we demonstrate via the simple yet effective baseline AddON that auxiliary data *can* improve OSR performance at any scale and regardless of auxiliary data selection. Our findings invalidate current best practices in OSR, demonstrating that small-scale evaluations with auxiliary data do not translate to large-scale performance. RL methods considered state-of-the-art based on CIFAR benchmarks, such as Outlier Exposure, can suffer from significant performance degradation below baselines when deployed at scale.

## 7 ETHICS STATEMENT

We do not foresee any ethical concerns with our work.

## 8 REPRODUCIBILITY STATEMENT

We provide a detailed description of our experimental setup in Section 4 and in-depth descriptions of the used methods and applied evaluation metrics in the appendix, alongside extensive results. We will open-source our modular code package used for this work upon publication and provide all scripts and parameter settings to facilitate reproducibility of all plots and tables provided within this work. The code package also allows for expansion to further research, *e.g.*, the inclusion of additional datasets and RL or PP methods.

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

# A APPENDIX

## A.1 METHOD SELECTION AND BACKGROUND

An overall view of the two-stage processing pipeline, including the nomenclature as used in this section, is given in Figure 8. Below, we provide details for the RL and PP methods that we investigate in this work as well as a discussion on method selection to answer our research questions.

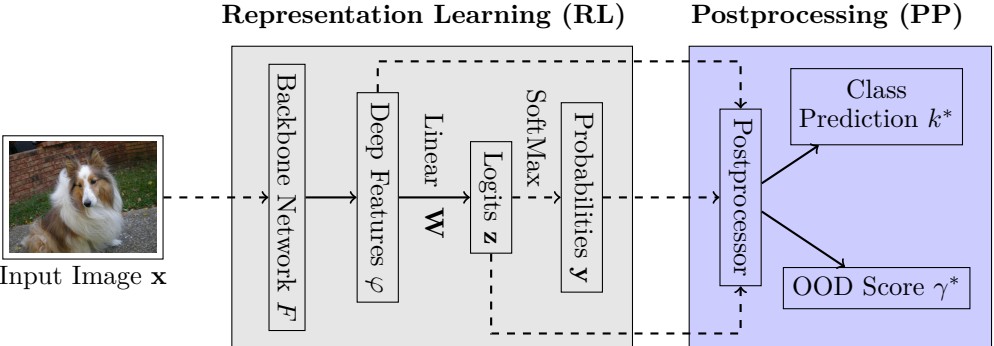

Figure 8: Two-stage processing framework for OSR. An image is presented to the backbone network $F$, which extracts deep features $\varphi$ that are then processed with a linear layer $\mathbf{W}$ to logits $\mathbf{z}$, and further with SoftMax to probabilities $\mathbf{y}$. Solid lines indicate (potentially) learnable connections, while dashed lines highlight non-learnable connections. The postprocessor takes the deep features, logits, or probabilities as input and outputs a class prediction $k^*$ and a score $\gamma^*$.

### A.1.1 REPRESENTATION LEARNING METHODS

**Methods Selection** Our model selection for RL aims to cover a representative set of methods that reflect the current state of the art in OSR as well as OOD detection while simultaneously adhering to a set of constraints given by our rigorous experimental setup to ensure that we can adequately test our research questions and hypotheses. We selected RL methods that satisfy the following criteria:

1. We require methods that utilize no or only natural auxiliary samples as this is at the core of our analysis.

2. Our modularization of OSR methods requires classification-based (discriminative) RL methods since most PP methods rely on output logits or probabilities to perform classification.

3. RL methods must be able to be trained from scratch on the ImageNet protocols without pre-training in order to prevent information leakage from training on classes that our protocols declare as unknown.

Following, we discuss each method and its formalization in the two-stage framework in more detail. For a systematic overview including method characteristics, see Table 1a.

**CE** Our baseline (Hendrycks & Gimpel, 2017) training-based approach is the categorical Cross-Entropy (CE) loss trained only on samples from $K$ known classes. For an input sample $(\mathbf{x}_n, \tau_n) \in \mathcal{K}$ and an arbitrary backbone network we obtain the deep features $\varphi_n \in \mathbb{R}^D$ for some deep feature dimension $D$. These features are then passed through a fully-connected *logit layer* $\mathbf{W} \in \mathbb{R}^{C \times D}$ with $C = K$ output *logits* $\mathbf{z}_n = \mathbf{W}\varphi_n \in \mathbb{R}^C$.[4] The logits are then

---

[4]Note that we can express the logit for class $c$ through the angle of the feature to the class center $\mathbf{W}_c$ as $\mathbf{z}_{n,c} = \mathbf{W}\varphi_n = \|\varphi_n\| \|W_c\| \cos(\alpha)$, where $\mathbf{W}_c$ is the $c$-th row vector of $\mathbf{W}$ and $\alpha$ is the angle between $\varphi_n$ and $\mathbf{W}_c$.

turned into probabilities $\mathbf{y}_n \in \mathbb{R}^K$ through SoftMax activation:

$$y_{n,c} = \frac{e^{z_{n,c}}}{\sum\limits_{c'=1}^{C} e^{z_{n,c'}}} \, . \tag{5}$$

Based on these, the CE loss is computed as:

$$\mathcal{J}_{\mathrm{CE}} = -\mathbb{E}_{(\mathbf{x}_n, \tau_n) \in \mathcal{K}} \log y_{n,\tau_n} \, . \tag{6}$$

**OE**  We include Outlier Exposure (OE) (Hendrycks et al., 2019) as state-of-the-art RL method from the OOD detection literature that utilize auxiliary data (Wang et al., 2025). OE adds a regularization term to (6) that maximizes the entropy for auxiliary samples by computing the CE loss between the uniform distribution and the SoftMax confidences of the network:

$$\mathcal{J}_{\mathrm{OE}} = \mathcal{J}_{\mathrm{CE}} - \lambda_{\mathrm{OE}} \, \mathbb{E}_{(\mathbf{x}_n, \tau_n) \in \mathcal{A}} \, \frac{1}{C} \sum_{c=1}^{C} \log y_{n,c} \tag{7}$$

OE essentially is equivalent to the Entropic Open-Set (EOS) loss $\mathcal{J}_{\mathrm{EOS}}$ proposed by Dhamija et al. (2018), with the only exception that OE provides a more intuitive way to weight the impact of auxiliary samples to the training (with $\lambda_{\mathrm{OE}} = 0.5$ for computer vision tasks), whereas EOS does so via class weighting (set to 1). From Dhamija et al. (2018) we know that EOS and, by extension, OE *implicitly* manipulate feature magnitudes, by encouraging the network to learn small feature magnitudes for auxiliary samples and large magnitudes for known samples.

**OS**  We employ the ObjectoSphere (OS) loss (Dhamija et al., 2018) as an extension to the EOS loss which *explicitly* manipulates feature magnitudes. It learns vanishing vectors for auxiliary samples and large magnitudes for known samples by using the following regularization term combined with the EOS loss:

$$\mathcal{J}_{\mathrm{OS}} = \mathcal{J}_{\mathrm{EOS}} + \lambda_{\mathrm{OS}} \begin{cases} \max(0, \xi - \|\varphi_n\|_2^2) & \text{if } \mathbf{x}_n \in \mathcal{K} \\ \|\varphi_n\|_2^2 & \text{if } \mathbf{x}_n \in \mathcal{A} \end{cases} \tag{8}$$

where the hyperparameter $\xi$ is the lower bound for the feature magnitude of known samples.

**AddON**  We use an RL method for OSR and OOD methods that utilize auxiliary data, which we term Additional Output Node (AddON). AddON uses the known data $\mathcal{K}$ as in CE, and auxiliary data $\mathcal{A}$ to train an additional output node $z_{n,K+1}$, so that we have a total of $C = K + 1$ outputs, creating a default class for all auxiliary and unknown samples. It is trained with the standard CE loss (6) with label $\tau = K + 1$ for auxiliary samples. While this is a common approach in object detection models (Dhamija et al., 2020), which collect a lot of background samples, it is only rarely applied directly to OSR problems (Dhamija et al., 2018; Palechor et al., 2023) or as component of more complex architectures such as PROSER or G-OpenMax (Zhou et al., 2021; Ge et al., 2017). Depending on the context, AddON is known as *Background Class* (Dhamija et al., 2020; 2018; Palechor et al., 2023), $K + 1$ (Kong & Ramanan, 2021), or *Dummy Classifier* (Zhou et al., 2021).

**ARPL**  Finally, we include the Adversarial Reciprocal Point Learning (ARPL) loss (Chen et al., 2021), which currently is the state of the art for OSR[5] and trained solely on the known data $\mathcal{K}$. Unlike most training-based methods, ARPL does not learn a prototype for a class $c$, but a reciprocal point $\mathbf{p}_c \in \mathbb{R}^D$ in deep feature space, *i.e.*, a point that represents everything *but* class $c$. The ARPL objective maximizes the distance between the reciprocal point and the features of samples from the respective class by computing logits $\mathbf{z}_n$ via distance measures between deep features $\varphi_n$ and the reciprocal points $\mathbf{p}_c$:

$$z_{n,c} = \frac{1}{D} \|\varphi_n - \mathbf{p}_c\|_2^2 - \varphi_n^{\mathrm{T}} \mathbf{p}_c \, , \tag{9}$$

---

[5]Note that we do not use ARPL+CS with the generator for confusing samples (CS), since it is prohibitively expensive to train at large scale (Vaze et al., 2022).

which are used via SoftMax (5) in the CE loss (6). To constrain open space, the distance between deep features and reciprocal points is bound by a learnable constant $\rho$ via the regularization term with weight $\lambda_{\text{ARPL}} = 0.1$:

$$\mathcal{J}_{\text{ARPL}} = \mathcal{J}_{\text{CE}} + \lambda_{\text{ARPL}} \mathbb{E}_{(\mathbf{x}_n, \tau_n) \in \mathcal{K}} \max\left(0, \frac{1}{D}\|\varphi_n - \mathbf{p}_{\tau_n}\|_2^2 - \rho\right) \tag{10}$$

### A.1.2 Postprocessing methods

**Method Selection**  We selected PP methods to cover a split between representative magnitude-aware and magnitude-unaware methods. Furthermore, we aim to cover both trainable and non-trainable methods to include methods that can specifically adjust to differently learned representations. Following, we discuss each method and its formalization in the two-stage framework in more detail. For a systematic overview including method characteristics, see Table 1b.

**MSP**  The Maximum Softmax Probability (MSP) (Hendrycks & Gimpel, 2017) is the de-facto default PP method for OSR and OOD detection and serves as our baseline. Class predictions and OOD scores are computed from probabilities for known classes as:[6]

$$k_n^* = \arg\max_{1 \leq c \leq K} y_{n,c} \qquad \text{and} \qquad \gamma_n^* = y_{n,k_n^*} \,. \tag{11}$$

MSP is not magnitude-aware since SoftMax (5) normalizes the logits and only considers their relative differences.

**MaxLogits**  MaxLogits (Hendrycks et al., 2022) or MLS (Vaze et al., 2022) exploit the magnitude of the logits $\mathbf{z}_n$, which contains useful information for OSR and OOD detection that is lost during softmax. MLS is magnitude-aware since logit magnitude is linked to feature magnitude (Wang et al., 2025).[4] Class predictions and OOD scores are computed from known logits as:

$$k_n^* = \arg\max_{1 \leq c \leq K} z_{n,c} \qquad \text{and} \qquad \gamma_n^* = z_{n,k_n^*} \,. \tag{12}$$

**OpenMax**  OpenMax (Bendale & Boult, 2016) uses deep features $\varphi_n$, referred to as Activation Vectors (AVs), to statistically model probabilities for an additional output node for the unknown class. During training, for each known class $c$, the Mean Activation Vector (MAV) $\mu_c$ is computed by averaging the deep features $\varphi_n$ extracted from all correctly classified training samples. Then, the cosine distances of the MAV $\mu_c$ to all the AVs $\varphi_{n,c}$ of the same class are computed. Here, we make use of a twist implemented in the VAST package of the original authors:[7] instead of using the original distances to model the distribution, we multiply the cosine distances by a factor $\kappa$, which allows modeling more compact class representations:

$$d_{n,c} = \kappa(1 - \cos(\varphi_n, \mu_c)) \tag{13}$$

Then, a per-class Weibull distribution $\Psi_c$ is fitted to the top $\lambda$ largest cosine distances $d_{n,c}$. For features $\varphi_n$ of a test sample, the class-wise Weibull distributions estimate a logit $\hat{z}_{n,K+1}$ for the unknown class, as well as modifying the logits for the top $\alpha$ classes, giving newly estimated logits $\hat{z}_{n,c}$. From these, the output $\hat{y}_{n,c}$ is computed[6] via softmax (5) and the output $\mathcal{P}_n$ is computed as:

$$k_n^* = \arg\max_{1 \leq c \leq K} \hat{z}_{n,c} \qquad \text{and} \qquad \gamma_n^* = \hat{y}_{n,k_n^*} \,. \tag{14}$$

OpenMax is not magnitude-aware since the cosine distance ignores the feature magnitude.

---

[6]Please note that for computing OOD scores $\gamma_n^*$, we purposefully ignore the unknown class ($y_{n,K+1}$ or $z_{n,K+1}$) if it exists. A low probability for the unknown class $y_{n,K+1}$ does not indicate a high probability for any of the known classes. On the other hand, due to Softmax requiring probabilities sum up to 1, a large probability $y_{n,K+1}$ enforces low probabilities for all known classes. Therefore, $y_{n,K+1}$ does not add any new information.

[7]https://github.com/Vastlab/vast

**PostMax**   Postnormalization of Maxima (PostMax) (Cruz et al., 2024) uses Extreme Value Theory (EVT) by applying a Generalized Pareto Distribution (GPD) to maximum logits post-normalized by the feature magnitude. Based on their findings that unknown samples have larger feature magnitude than known samples on large-scale data, they normalize logits by dividing them by their deep feature magnitude to further increase the separation between known and unknown samples:

$$\hat{\mathbf{z}}_n = \frac{\mathbf{z}_n}{\|\varphi_n\|_2 + 1}.$$ (15)

This makes it an explicitly magnitude-aware method. We modify the original implementation to shift the magnitudes by 1 to avoid issues with magnitudes $\|\varphi\|_2 < 1$ which reverse the desired effect of normalization.[8] This occurs consistently for features from magnitude-manipulating RL methods, but not for others. The class-agnostic GPD $\Psi_{\mu,\sigma,\xi}$ is fitted only on the maximum normalized logits of correctly classified known training samples, which is then used to compute a probability of the sample being known. Following Cruz et al. (2024), the scores $\mathcal{P}_n$ are computed as:

$$k_n^* = \underset{1 \leq c \leq K}{\arg\max}\, \hat{z}_{n,c} \qquad \text{and} \qquad \gamma_n^* = \Psi_{\mu,\sigma,\xi}(\hat{z}_{n,k_n^*}).$$ (16)

**GHOST**   The Gaussian Hypothesis Open-Set Technique (GHOST) (Rabinowitz et al., 2025) models each class in deep feature space as a multivariate Gaussian distribution with the intuition that features $\varphi$ from known and unknown samples deviate by feature magnitude even if the angular direction overlaps, making it magnitude-aware. During training, GHOST fits a Gaussian distribution $(\mu_c, \sigma_c)$ for each known class $c$ based on the deep features $\varphi_n$ of correctly classified training samples. During inference, GHOST first computes a class prediction from the logits:

$$k_n^* = \underset{1 \leq c \leq K}{\arg\max}\, z_{n,c},$$ (17)

as well as z-score $s_n$ for each sample $\mathbf{x}_n$ and corresponding Gaussian $(\mu_{k_n^*}, \sigma_{k_n^*})$. This is then used to compute the score $\gamma_n^*$ by dividing the original logit:

$$s_n = \sum_{d=1}^{D} \frac{|\varphi_{n,d} - \mu_{k_n^*,d}|}{\sigma_{k_n^*,d}}, \qquad \gamma_n^* = \frac{z_{n,k_n^*}}{s_n}.$$ (18)

## A.2   DERIVATION FOR FEATURE MAGNITUDE INCENTIVE

Here, we provide the derivation for the training incentive of CE with $C = K$ (or AddON with $C = K + 1$) to learn "sufficiently large" feature magnitudes:

$$\|\varphi\|_2 > \frac{l}{\delta_{\mathbf{z}_c/\|\varphi\|_2}} = \frac{\log(\frac{1-\epsilon}{\epsilon}) + \log(C-1)}{\|\mathbf{W}_c\|_2 \cos(\alpha_c) - \max_{c' \neq c}\{\|\mathbf{W}_{c'}\|_2 \cos(\alpha_{c'})\}} \implies \mathbf{y}_c > 1 - \epsilon,$$

For some small $\epsilon > 0$ and class $c \in \{1, \ldots, C\}$, we want the trained network to be able to achieve the maximum probability over all classes $\mathbf{y}_c > 1 - \epsilon$ for any sample. Note that we

---

[8]We also tried different normalization techniques, including to multiply with the norm, which seems more reasonable for MM RL methods. However, the detrimental effects of MM RL with PostMax for large-scale evaluations was present in any case. This modification does not harm performance with non-MM RL methods.

have $\mathbf{y}_c = \max_{c' \in \{1,\dots,C\}} \mathbf{y}_{c'}$.

$$\mathbf{y}_c > 1 - \epsilon \iff \frac{e^{\mathbf{z}_c}}{\sum_{c'} e^{\mathbf{z}_{c'}}} = \frac{e^{\mathbf{z}_c}}{e^{\mathbf{z}_c} + \sum_{c' \neq c} e^{\mathbf{z}_{c'}}} > 1 - \epsilon$$

$$\iff e^{\mathbf{z}_c} > (1 - \epsilon)(e^{\mathbf{z}_c} + \sum_{c' \neq c} e^{\mathbf{z}_{c'}})$$

$$\iff e^{\mathbf{z}_c} > e^{\mathbf{z}_c} + \sum_{c' \neq c} e^{\mathbf{z}_{c'}} + \epsilon e^{\mathbf{z}_c} - \epsilon \sum_{c' \neq c} e^{\mathbf{z}_{c'}}$$

$$\iff e^{\mathbf{z}_c} > \frac{1 - \epsilon}{\epsilon} \sum_{c' \neq c} e^{\mathbf{z}_{c'}}$$

$$\iff \mathbf{z}_c > \underbrace{\log\left(\frac{1 - \epsilon}{\epsilon}\right)}_{=:\beta} + \log\left(\sum_{c' \neq c} e^{\mathbf{z}_{c'}}\right)$$

By binding the LogSumExp $\log\left(\sum_{c' \neq c} e^{\mathbf{z}_{c'}}\right)$ above, we get a sufficient condition $\delta_{\mathbf{z}_c} > l$ such that $\delta_{\mathbf{z}_c} > l \implies \mathbf{y}_c > 1 - \epsilon$. We have the upper bound of the LogSumExp function:

$$\log\left(\sum_{c' \neq c} e^{\mathbf{z}_{c'}}\right) \leq \log(C - 1) + \max_{c \neq c'} \mathbf{z}_{c'}$$

$$\mathbf{z}_c > \beta + \max_{c \neq c'} \mathbf{z}_{c'} + \log(C - 1) \implies \mathbf{z}_c > \beta + \log\left(\sum_{c' \neq c} e^{\mathbf{z}_{c'}}\right)$$

From this we get:

$$\mathbf{z}_c > \beta + \max_{c \neq c'} \mathbf{z}_{c'} + \log(C - 1) \iff \underbrace{\mathbf{z}_c - \max_{c \neq c'} \mathbf{z}_{c'}}_{=:\delta_{\mathbf{z}_c}} > \beta + \log(C - 1) =: l$$

This provides a lower bound on the difference between the maximal two logits. Using footnote 4 we have

$$\iff \|\varphi\|_2 \|\mathbf{W}_c\|_2 \cos(\alpha_{\varphi, \mathbf{W}_c}) - \max_{c' \neq c}\{\|\varphi\|_2 \|\mathbf{W}_{c'}\|_2 \cos(\alpha_{\varphi, \mathbf{W}_{c'}})\} > l$$

$$\iff \|\varphi\|_2 \underbrace{(\|\mathbf{W}_c\|_2 \cos(\alpha_{\varphi, \mathbf{W}_c}) - \max_{c' \neq c} \|\mathbf{W}_{c'}\|_2 \cos(\alpha_{\varphi, \mathbf{W}_{c'}})\})}_{=:\delta_{\mathbf{z}_c / \|\varphi\|_2}} > l$$

$$\iff \|\varphi\|_2 > \frac{l}{\delta_{\mathbf{z}_c / \|\varphi\|_2}}$$

where $\delta_{\mathbf{z}_c / \|\varphi\|_2} > 0$.

## A.3 EMNIST BENCHMARK

We perform qualitative small-scale experiments on the **EMNIST** protocol (Dhamija et al., 2018) to replicate settings of high similarity between known and auxiliary classes at small scale. This benchmark contains a wide spectrum of clearly attributable visual similarities between known and auxiliary classes because the samples do not contain any image background. It consists of EMNIST MNIST split as knowns, the first half of EMNIST letters (a-m) as auxiliary data, and the second half (n-z) as unknowns.[9] While most digits (known)

---

[9]Contrary to (Dhamija et al., 2018) we use EMNIST MNIST instead of the original MNIST because the preprocessing, while similar, is not exact (Cohen et al., 2017). EMNIST MNIST and letters contain noticeably softer and thicker digits than MNIST. In order to focus on semantic shift and not introduce easily learnable covariate shift through softness and sharpness, we use EMNIST MNIST.

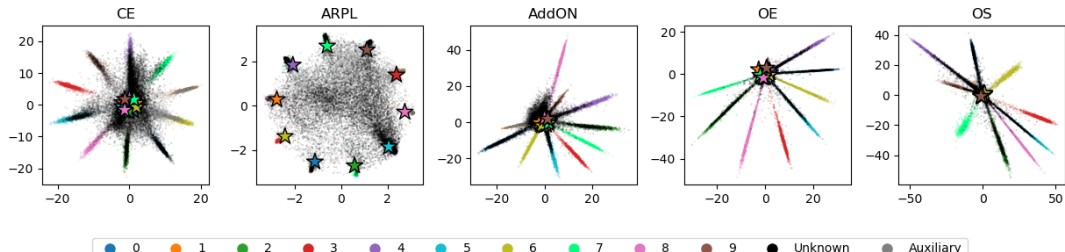

Figure 9: 2D feature visualizations of EMNIST experiments with different RL methods.

| Method | Parameter | CIFAR+N | | ImageNet | | |
| | | CIFAR+10 | CIFAR+50 | $P_1$ | $P_2$ | $P_3$ |
| --- | --- | --- | --- | --- | --- | --- |
| OS | $\lambda_{OS}$ | $10^{-3}/10^{-2}/10^{-2}/10^{-2}/10^{-2}$ | $10^{-3}/10^{-2}/10^{-2}/10^{-3}/10^{-2}$ | $10^{-3}$ | $10^{-3}$ | $10^{-3}$ |
| | $\xi$ | 10/10/10/10/10 | 20/10/10/10/10 | 30 | 20 | 10 |
| CE+OpenMax | $\alpha$ | 3/3/3/1/3 | 3/2/3/3/3 | 5 | 5 | 2 |
| | $\kappa$ | 1.5/1.5/2.3/2/1.7 | 1.7/1.5/2.3/1.5/2.3 | 2.3 | 2.3 | 2 |
| | $\lambda$ | 100/100/250/10/100 | 250/100/500/100/100 | 750 | 750 | 100 |
| ARPL+OpenMax | $\alpha$ | 3/3/1/3/3 | 3/2/3/3/3 | 5 | 3 | 2 |
| | $\kappa$ | 1.5/2.3/2/1.5/1.7 | 2/1.5/1.5/1.5/1.5 | 2.3 | 2.3 | 2.3 |
| | $\lambda$ | 100/250/10/100/100 | 100/100/100/100/100 | 750 | 750 | 10 |
| AddON+OpenMax | $\alpha$ | 1/1/1/1/1 | 1/1/1/1/1 | 5 | 3 | 5 |
| | $\kappa$ | 1.5/2/2.3/1.7/2 | 1.7/2/1.5/1.5/2.3 | 1.7 | 2.3 | 2 |
| | $\lambda$ | 100/250/100/100/100 | 250/500/250/250/250 | 750 | 750 | 100 |
| OE+OpenMax | $\alpha$ | 1/2/1/3/3 | 3/3/1/2/1 | 3 | 2 | 2 |
| | $\kappa$ | 1.5/1.7/2/2/2 | 2.3/1.5/1.5/2/1.5 | 1.7 | 1.5 | 2.3 |
| | $\lambda$ | 10/100/10/100/250 | 100/10/10/100/10 | 10 | 10 | 10 |
| OS+OpenMax | $\alpha$ | 1/1/3/2/2 | 3/2/3/3/2 | 10 | 5 | 10 |
| | $\kappa$ | 1.5/2.3/1.5/1.5/2.3 | 2.3/1.7/2/2.3/2 | 2.3 | 2.3 | 2 |
| | $\lambda$ | 10/10/10/10/10 | 100/10/10/100/10 | 10 | 10 | 10 |

Table 2: Optimized hyperparameter values for each method and protocol. CIFAR+N results are reported for each trial separately.

do not contain any visually similar letters, digits 1 and 9 contain auxiliary classes ("i", "l" and "g", "q" respectively) that are visually very similar or indistinguishable without context depending on handwriting and capitalization.

To visualize how magnitude collapse and known class classification interacts, we repeat the EMNIST experiments with a 2D bottleneck deep feature layer, following Dhamija et al. (2018). These can illustrate how different RL methods structure the feature space and how this affects PP methods, however, the low dimensionality limits not only the representational capacity but also changes the geometry of the feature space and the behavior, so extrapolation to high dimensional feature spaces is limited. Experiments are otherwise identical to those in Section 4.

## A.4 HYPERPARAMETER OPTIMIZATION

For most methods, we rely on the hyperparameters as provided by the according papers. Few methods however do not provide any, namely: $\lambda$ and $\xi$ for ObjectoSphere and $\lambda$, $\kappa$, and $\alpha$ for OpenMax, where we perform hyperparameter optimization on the validation set using grid search, based on the maximum AUOSCR. For OpenMax the parameter optimization is performed for each upstream RL method separately to ensure optimal settings, since the feature representations differ significantly. The considered hyperparameter ranges are as follows:

- OS: $\lambda \in \{10^{-4}, 10^{-3}, 10^{-2}\}$, $\xi \in \{10, 20, 30\}$
- OpenMax: $\alpha \in \{1, 2, 3, 5, 10\}$, $\kappa \in \{1.5, 1.7, 2, 2.3\}$, $\lambda \in \{10, 50, 100, 250, 500, 750, 1000\}$

The best hyperparameter values for each method and protocol are summarized in Table 2.

### A.5 Performance metrics

**CCR@FPR** CCR@FPR computes the CCR at a specific FPRs $\zeta \in \{10^{-4}, 10^{-3}, 10^{-2}, 10^{-1}, 10^0\}$ and provides insights into the classification performance at various tolerances for errors caused by missed rejections. It is therefore highly relevant for practical applications where a fixed threshold $\theta$ is required, which is typically selected based on a certain FPR. It is computed as:

$$\text{CCR@FPR} = \begin{cases} \text{CCR}(\theta_\zeta) & \text{if } \theta_\zeta \text{ exists} \\ 0 & \text{otherwise} \end{cases}, \tag{19}$$

where $\theta_\zeta = \text{FPR}^{-1}(\zeta)$ is the threshold that yields $\text{FPR} = \zeta$. CCR@FPR for $\zeta = 10^0$ resembles closed-set accuracy.

**AUOSCR** The OSCR curve (Dhamija et al., 2018) simultaneously evaluates classification of known samples via CCR as well as the rejection of unknown samples via FPR over all possible thresholds. It is computed by varying the threshold $\theta$ from the smallest to the largest possible score value, and plotting the CCR over the FPR, *i.e.*, computing CCR@FPR at all thresholds. The Area Under the OSCR curve (AUOSCR) is computed by integrating the OSCR curve from $\zeta = 0$ to $\zeta = 1$. Since the OSCR curve is a monotonically increasing function, the AUOSCR is maximized at and bounded by the closed-set accuracy.

**OOSA** The Operational Open-set Accuracy (OOSA) (Cruz et al., 2024) evaluates the open-set performance at a fixed operational threshold $\theta^\star$, determined on the validation set, and provides insights into the performance of a method in a real-world setting. It is defined as a trade-off between the CCR and the Unknown Rejection Rate (URR), $\text{URR}(\theta) = 1 - \text{FPR}(\theta)$:

$$\text{OOSA} = \alpha_{\text{CCR}}\text{CCR}(\theta^\star) + (1 - \alpha_{\text{CCR}})\text{URR}(\theta^\star) \tag{20}$$

where $\theta^\star$ is the operational threshold that maximizes this equation on the validation set. We follow Cruz et al. (2024) and set $\alpha_{\text{CCR}} = \frac{|\mathcal{K}_{\text{test}}|}{|\mathcal{K}_{\text{test}}| + |\mathcal{U}_{\text{test}}|}$ to equally weight known and unknown test samples.

**AUROC** In order to evaluate the OOD detection capabilities independently of the ID classification task, we use the Area Under the Receiver Operating Characteristics (AUROC) curve (Hendrycks & Gimpel, 2017; Hendrycks et al., 2019; Chen et al., 2021; Hendrycks et al., 2022; Vaze et al., 2022; Yang et al., 2024; Wang et al., 2025). AUROC concerns how well known and unknown classes can be distinguished by computing FPR (2), as well as the True Positive Rate (TPR):

$$\text{TPR}(\theta) = \frac{\left|\{(\mathbf{x}_n, \tau_n) \in \mathcal{K} \wedge \gamma_n^* \geq \theta\}\right|}{N_K} \tag{21}$$

The ROC is computed by varying $\theta$, and the area under that curve is determined.

### A.6 Additional figures and tables

The main paper contained only a subset of evaluation metrics and visualizatrions. Here we provide remaining figures and tables containing the exact results.

Table 3: Small-scale evaluation. This table includes a performance overview of all RL and PP methods on the small-scale protocols. Metrics include CCR@$\zeta$ for $\zeta \in 10^{-2}, 10^{-1}, 10^{-0}, 10^{1}, 10^{2}$, AUOSCR, AUROC, and OOSA. All scores are reported in percent and CCR@100 is the closed-set accuracy. Performance metrics are reported as mean $\pm$ standard deviation over 5 randomized trials. The best performing combination for each protocol and metric w.r.t mean score is highlighted in bold. The best performing PP method for each RL method and metric w.r.t mean score is highlighted in italic.

| Dataset | RL | PP | CCR@0.01 | CCR@0.1 | CCR@1 | CCR@10 | CCR@100 | AUOSCR | AUROC | OOSA |
|---|---|---|---|---|---|---|---|---|---|---|
| CIFAR+10 | CE | MSP | 43.0 ± 13.2 | 40.5 ± 13.3 | 63.8 ± 5.1 | 84.5 ± 2.4 | 97.0 ± 0.7 | 92.2 ± 1.1 | 93.7 ± 0.8 | 87.8 ± 1.1 |
| | | OpenMax | 43.8 ± 12.6 | 44.1 ± 12.3 | 67.8 ± 3.0 | 86.1 ± 1.9 | 97.0 ± 0.7 | 92.9 ± 0.9 | 94.5 ± 0.5 | 88.4 ± 1.0 |
| | | MaxLogits | *46.5 ± 9.2* | *46.5 ± 9.2* | *72.5 ± 2.7* | *89.8 ± 1.7* | 97.0 ± 0.7 | *94.4 ± 0.9* | *96.6 ± 0.5* | *90.5 ± 1.0* |
| | | PostMax | 33.6 ± 15.3 | 33.6 ± 15.3 | 64.6 ± 4.3 | 85.3 ± 1.6 | 97.0 ± 0.7 | 92.8 ± 0.7 | 94.7 ± 0.3 | 88.3 ± 1.0 |
| | | GHOST | 3.2 ± 3.7 | 2.6 ± 3.5 | 28.1 ± 10.2 | 87.7 ± 1.3 | 97.0 ± 0.7 | 93.0 ± 0.7 | 95.1 ± 0.3 | 90.0 ± 1.1 |
| | ARPL | MSP | 41.0 ± 11.8 | 42.8 ± 9.6 | 65.3 ± 1.7 | 84.6 ± 2.1 | 97.1 ± 0.7 | 92.4 ± 1.1 | 93.9 ± 0.8 | 87.9 ± 1.1 |
| | | OpenMax | 44.4 ± 9.4 | 44.6 ± 9.2 | 68.2 ± 1.5 | 86.4 ± 1.6 | 97.1 ± 0.7 | 93.0 ± 0.9 | 94.7 ± 0.5 | 88.6 ± 0.9 |
| | | MaxLogits | *48.8 ± 12.4* | *48.8 ± 12.4* | *73.2 ± 2.9* | *90.2 ± 1.2* | 97.1 ± 0.7 | *94.5 ± 0.8* | *96.8 ± 0.4* | *90.7 ± 1.0* |
| | | PostMax | 36.3 ± 15.0 | 36.3 ± 15.0 | 64.3 ± 2.4 | 85.5 ± 1.4 | 97.1 ± 0.7 | 93.0 ± 0.8 | 94.9 ± 0.4 | 88.7 ± 0.7 |
| | | GHOST | 3.6 ± 4.3 | 3.6 ± 4.3 | 23.3 ± 7.1 | 88.1 ± 1.4 | 97.1 ± 0.7 | 93.0 ± 0.8 | 95.2 ± 0.4 | 90.2 ± 1.0 |
| | AddON | MSP | 58.0 ± 18.2 | 58.1 ± 18.2 | 80.7 ± 7.6 | 92.6 ± 2.5 | 97.3 ± 0.7 | 95.5 ± 0.9 | 97.9 ± 1.4 | 92.2 ± 2.5 |
| | | OpenMax | 61.8 ± 12.9 | 61.8 ± 12.9 | 80.8 ± 4.0 | 92.0 ± 1.6 | **97.4 ± 0.7** | 95.4 ± 0.7 | 97.3 ± 0.9 | 91.9 ± 1.2 |
| | | MaxLogits | *63.1 ± 22.8* | *63.1 ± 22.8* | ***84.2 ± 4.4*** | ***94.2 ± 0.9*** | 97.3 ± 0.7 | ***96.2 ± 0.4*** | ***98.6 ± 0.6*** | ***93.7 ± 0.7*** |
| | | PostMax | 57.9 ± 19.8 | 57.9 ± 19.8 | 81.3 ± 7.8 | 93.5 ± 1.9 | 97.3 ± 0.7 | 95.8 ± 0.6 | 98.3 ± 1.0 | 93.0 ± 1.8 |
| | | GHOST | 34.8 ± 31.4 | 34.8 ± 31.4 | 81.1 ± 7.7 | 94.1 ± 1.3 | 97.3 ± 0.7 | 96.0 ± 0.5 | 98.4 ± 0.8 | 93.6 ± 1.1 |
| | OE | MSP | 57.4 ± 17.4 | 57.4 ± 17.3 | 81.4 ± 6.1 | 92.9 ± 1.8 | 97.1 ± 0.8 | 95.6 ± 0.6 | 98.0 ± 1.0 | 92.6 ± 1.6 |
| | | OpenMax | 58.2 ± 18.2 | 58.2 ± 18.2 | 81.5 ± 5.7 | 93.2 ± 1.8 | *97.2 ± 0.6* | *95.7 ± 0.6* | 98.0 ± 0.9 | 92.7 ± 1.5 |
| | | MaxLogits | *58.8 ± 19.5* | *58.8 ± 19.5* | *82.4 ± 5.8* | *93.3 ± 1.5* | 97.1 ± 0.8 | 95.6 ± 0.6 | *98.2 ± 0.9* | 92.6 ± 1.6 |
| | | PostMax | 52.3 ± 19.6 | 52.3 ± 19.6 | 81.3 ± 6.2 | 93.2 ± 1.5 | 97.1 ± 0.8 | 95.6 ± 0.6 | 98.1 ± 0.9 | *92.8 ± 1.5* |
| | | GHOST | 36.0 ± 32.5 | 36.0 ± 32.5 | 81.7 ± 6.4 | *93.3 ± 1.5* | 97.1 ± 0.8 | 95.6 ± 0.6 | 98.1 ± 1.0 | 92.7 ± 1.6 |
| | OS | MSP | ***65.5 ± 14.8*** | ***65.5 ± 14.8*** | 82.1 ± 6.5 | 93.2 ± 2.0 | **97.4 ± 0.7** | 95.8 ± 0.8 | 98.0 ± 1.1 | *92.7 ± 1.9* |
| | | OpenMax | 65.0 ± 17.7 | 65.0 ± 17.7 | 81.9 ± 6.9 | 93.3 ± 2.0 | 97.3 ± 0.8 | 95.8 ± 0.9 | 98.1 ± 1.1 | *92.7 ± 1.7* |
| | | MaxLogits | 65.2 ± 19.4 | 65.2 ± 19.4 | *82.9 ± 6.6* | *93.6 ± 1.8* | **97.4 ± 0.7** | *95.9 ± 0.7* | *98.3 ± 1.0* | *92.7 ± 1.8* |
| | | PostMax | 44.0 ± 18.2 | 44.0 ± 18.2 | 72.1 ± 10.2 | 92.6 ± 1.8 | **97.4 ± 0.7** | 95.4 ± 0.9 | 97.5 ± 1.0 | 92.5 ± 1.1 |
| | | GHOST | 42.0 ± 33.9 | 42.0 ± 33.9 | 78.8 ± 10.0 | 93.5 ± 1.8 | **97.4 ± 0.7** | 95.7 ± 0.8 | 98.1 ± 1.2 | 92.6 ± 2.0 |
| CIFAR+50 | CE | MSP | 22.9 ± 0.0 | 31.9 ± 3.7 | 54.2 ± 3.9 | 79.4 ± 1.6 | 97.1 ± 0.7 | 90.4 ± 0.7 | 91.6 ± 0.6 | 84.4 ± 0.7 |
| | | OpenMax | *24.7 ± 7.2* | 35.4 ± 5.3 | 58.4 ± 3.4 | 81.8 ± 1.3 | 97.1 ± 0.7 | 91.2 ± 0.6 | 92.7 ± 0.5 | 85.5 ± 0.8 |
| | | MaxLogits | 20.4 ± 6.3 | *38.0 ± 6.3* | *61.1 ± 2.8* | *84.3 ± 1.0* | 97.1 ± 0.7 | *92.5 ± 0.4* | *94.5 ± 0.4* | *86.0 ± 0.7* |
| | | PostMax | 11.4 ± 1.4 | 24.2 ± 6.8 | 51.8 ± 5.1 | 79.5 ± 1.2 | 97.1 ± 0.7 | 90.7 ± 0.4 | 92.3 ± 0.4 | 84.0 ± 0.9 |
| | | GHOST | 0.3 ± 0.2 | 1.4 ± 0.9 | 11.6 ± 4.2 | 80.8 ± 0.7 | 97.1 ± 0.7 | 90.0 ± 0.3 | 91.8 ± 0.4 | 85.5 ± 0.6 |
| | ARPL | MSP | *30.0 ± 0.0* | 38.3 ± 3.3 | 55.3 ± 5.3 | 80.3 ± 1.6 | 97.2 ± 0.6 | 90.6 ± 0.9 | 91.8 ± 0.6 | 85.0 ± 0.9 |
| | | OpenMax | 23.3 ± 8.3 | 36.1 ± 8.2 | 58.5 ± 5.0 | 82.1 ± 1.3 | 97.1 ± 0.5 | 91.3 ± 0.8 | 92.8 ± 0.5 | 85.2 ± 1.2 |
| | | MaxLogits | 23.5 ± 12.8 | *36.6 ± 6.6* | *62.1 ± 4.7* | *85.0 ± 1.0* | 97.2 ± 0.6 | *92.7 ± 0.6* | *94.7 ± 0.4* | *86.4 ± 0.8* |
| | | PostMax | 10.3 ± 5.0 | 23.1 ± 7.2 | 51.8 ± 4.0 | 80.1 ± 1.3 | 97.2 ± 0.6 | 90.9 ± 0.6 | 92.5 ± 0.4 | 84.9 ± 0.7 |
| | | GHOST | 0.1 ± 0.2 | 1.2 ± 0.9 | 12.0 ± 5.6 | 81.3 ± 1.4 | 97.2 ± 0.6 | 90.2 ± 0.6 | 92.0 ± 0.5 | 86.0 ± 0.6 |
| | AddON | MSP | *48.9 ± 12.6* | *65.4 ± 4.4* | *83.8 ± 2.9* | 93.8 ± 0.9 | **97.7 ± 0.7** | *96.3 ± 0.4* | ***98.4 ± 0.5*** | *93.6 ± 1.0* |
| | | OpenMax | 40.4 ± 6.7 | 60.8 ± 6.4 | 79.3 ± 2.9 | 91.6 ± 1.2 | 97.4 ± 0.7 | 95.2 ± 0.7 | 96.9 ± 0.4 | 91.8 ± 1.0 |
| | | MaxLogits | 38.9 ± 6.6 | 60.2 ± 6.9 | 79.8 ± 2.3 | 93.0 ± 0.5 | **97.7 ± 0.7** | 95.9 ± 0.4 | 97.8 ± 0.4 | 92.3 ± 0.6 |
| | | PostMax | 41.9 ± 17.6 | 60.5 ± 4.2 | 81.8 ± 3.0 | *93.9 ± 0.7* | **97.7 ± 0.7** | 96.2 ± 0.4 | 98.1 ± 0.5 | 93.1 ± 0.8 |
| | | GHOST | 5.5 ± 5.3 | 36.1 ± 14.8 | 80.8 ± 2.7 | 93.5 ± 0.6 | **97.7 ± 0.7** | 96.0 ± 0.5 | 97.9 ± 0.4 | 92.8 ± 0.5 |
| | OE | MSP | 50.1 ± 14.4 | 66.7 ± 7.6 | 83.5 ± 3.4 | 94.0 ± 1.1 | 97.6 ± 0.4 | 96.2 ± 0.3 | 98.2 ± 0.6 | 93.4 ± 1.0 |
| | | OpenMax | 49.6 ± 13.1 | *67.8 ± 6.7* | 83.5 ± 3.5 | 94.1 ± 1.0 | ***97.7 ± 0.5*** | ***96.4 ± 0.3*** | 98.2 ± 0.5 | *93.6 ± 0.9* |
| | | MaxLogits | *50.4 ± 14.3* | 67.1 ± 7.7 | *83.7 ± 3.7* | ***94.2 ± 1.0*** | 97.6 ± 0.4 | 96.2 ± 0.3 | *98.3 ± 0.6* | *93.6 ± 1.0* |
| | | PostMax | 46.8 ± 17.4 | 67.0 ± 6.8 | 83.2 ± 3.5 | 94.0 ± 1.1 | 97.6 ± 0.4 | 96.2 ± 0.3 | 98.2 ± 0.6 | 93.4 ± 0.9 |
| | | GHOST | 7.7 ± 5.2 | 46.8 ± 21.1 | 83.4 ± 3.6 | 94.1 ± 1.0 | 97.6 ± 0.4 | 96.2 ± 0.3 | 98.2 ± 0.6 | *93.6 ± 0.9* |
| | OS | MSP | 51.2 ± 11.8 | 70.2 ± 8.2 | 85.1 ± 2.5 | 94.0 ± 1.1 | *97.4 ± 0.5* | *96.2 ± 0.4* | *98.3 ± 0.5* | 93.9 ± 0.8 |
| | | OpenMax | 48.0 ± 11.8 | 70.1 ± 8.8 | 85.1 ± 2.6 | 94.0 ± 1.1 | *97.4 ± 0.5* | *96.2 ± 0.4* | *98.3 ± 0.5* | 93.8 ± 0.8 |
| | | MaxLogits | *51.6 ± 11.9* | *70.4 ± 8.4* | ***85.5 ± 2.6*** | 94.1 ± 1.0 | *97.4 ± 0.5* | *96.2 ± 0.4* | *98.3 ± 0.5* | 93.9 ± 0.8 |
| | | PostMax | 35.9 ± 19.7 | 53.5 ± 8.1 | 78.2 ± 4.2 | 92.7 ± 1.4 | *97.4 ± 0.5* | 95.6 ± 0.6 | 97.5 ± 0.6 | 92.5 ± 0.7 |
| | | GHOST | 8.2 ± 9.9 | 56.8 ± 28.2 | ***85.5 ± 3.0*** | 94.1 ± 1.1 | *97.4 ± 0.5* | *96.2 ± 0.4* | *98.3 ± 0.6* | ***94.0 ± 0.8*** |

Table 4: Large-scale evaluation. This table repeats the performance overview of Table 3 on large-scale protocols. — indicates that at no threshold $\theta_\zeta$ for corresponding FPR $\zeta$ could be achieved.

| Dataset | RL | PP | CCR@0.01 | CCR@0.1 | CCR@1 | CCR@10 | CCR@100 | AUOSCR | AUROC | OOSA |
|---|---|---|---|---|---|---|---|---|---|---|
| $P_1$ | CE | MSP | 7.8 | 22.7 | 53.4 | 71.8 | 77.4 | 75.6 | 95.3 | 83.7 |
| | | OpenMax | 13.2 | 34.4 | 65.0 | 76.6 | 77.0 | 76.5 | 98.7 | 86.2 |
| | | MaxLogits | 6.1 | 31.3 | 64.1 | 76.6 | 77.4 | 76.8 | 98.5 | 86.3 |
| | | PostMax | 14.0 | 33.0 | 65.9 | 75.9 | 77.4 | 76.7 | 98.0 | 86.6 |
| | | GHOST | 18.3 | 44.9 | 71.4 | 77.0 | 77.4 | 77.1 | 99.1 | 87.5 |
| | ARPL | MSP | 10.6 | 27.2 | 54.8 | 72.5 | 78.2 | 76.4 | 95.6 | 83.6 |
| | | OpenMax | 8.0 | 44.6 | 66.1 | 77.0 | 77.3 | 76.8 | 98.8 | 86.1 |
| | | MaxLogits | 27.8 | 35.1 | 66.3 | 77.4 | 78.2 | 77.7 | 98.7 | 86.8 |
| | | PostMax | 20.6 | 41.5 | 65.6 | 76.8 | 78.2 | 77.5 | 98.1 | 86.5 |
| | | GHOST | 21.1 | 48.9 | 72.3 | 77.9 | 78.2 | 77.9 | 99.2 | 87.7 |
| | AddON | MSP | 11.6 | 32.2 | 62.2 | 77.4 | 78.4 | 77.5 | 97.9 | 79.0 |
| | | OpenMax | 16.2 | 41.5 | 65.5 | 77.8 | 78.3 | 77.7 | 98.7 | 87.4 |
| | | MaxLogits | 17.1 | 39.4 | 72.6 | 78.1 | 78.4 | 78.1 | 99.3 | 88.7 |
| | | PostMax | 14.5 | 35.9 | 67.3 | 77.0 | 78.4 | 77.7 | 98.4 | 86.9 |
| | | GHOST | 19.7 | 51.3 | 74.8 | 78.1 | 78.4 | 78.2 | 99.5 | 89.0 |
| | OE | MSP | 5.6 | 27.9 | 62.7 | 77.6 | 78.1 | 77.4 | 98.2 | 84.3 |
| | | OpenMax | 5.6 | 28.2 | 62.4 | 77.4 | 78.0 | 77.3 | 98.2 | 83.7 |
| | | MaxLogits | 16.1 | 36.7 | 73.1 | 77.7 | 78.1 | 77.8 | 99.0 | 84.3 |
| | | PostMax | 8.8 | 27.2 | 63.3 | 77.1 | 78.1 | 77.4 | 98.2 | 84.2 |
| | | GHOST | 12.1 | 48.0 | 74.4 | 77.8 | 78.1 | 77.8 | 99.1 | 86.4 |
| | OS | MSP | 18.0 | 34.2 | 64.9 | 77.8 | 78.3 | 77.7 | 98.6 | 83.4 |
| | | OpenMax | 22.5 | 33.4 | 64.6 | 77.6 | 78.0 | 77.4 | 98.6 | 84.3 |
| | | MaxLogits | 16.4 | 46.0 | 73.9 | 78.0 | 78.3 | 78.0 | 99.3 | 86.4 |
| | | PostMax | 21.4 | 41.4 | 70.7 | 78.0 | 78.3 | 77.9 | 99.1 | 87.3 |
| | | GHOST | 19.5 | 54.5 | 75.9 | 78.0 | 78.3 | 78.0 | 99.4 | 83.9 |
| $P_2$ | CE | MSP | — | — | 21.8 | 50.5 | 74.0 | 64.9 | 80.1 | 76.4 |
| | | OpenMax | 2.5 | 3.6 | 24.5 | 51.5 | 73.7 | 66.3 | 83.4 | 77.1 |
| | | MaxLogits | 7.6 | 8.4 | 25.0 | 58.9 | 74.0 | 68.5 | 87.8 | 78.8 |
| | | PostMax | 6.0 | 7.4 | 30.5 | 59.9 | 74.0 | 68.8 | 88.2 | 79.4 |
| | | GHOST | 8.1 | 10.3 | 30.3 | 61.3 | 74.0 | 69.1 | 88.9 | 80.0 |
| | ARPL | MSP | — | — | 24.4 | 49.9 | 75.4 | 66.1 | 80.7 | 76.3 |
| | | OpenMax | 2.6 | 4.1 | 29.5 | 52.2 | 73.8 | 66.7 | 83.6 | 77.2 |
| | | MaxLogits | 3.1 | 3.7 | 26.9 | 58.5 | 75.4 | 69.2 | 87.5 | 78.8 |
| | | PostMax | 5.7 | 13.6 | 34.5 | 59.8 | 75.4 | 69.8 | 88.3 | 80.2 |
| | | GHOST | 4.5 | 13.3 | 34.5 | 60.1 | 75.4 | 70.1 | 88.9 | 80.5 |
| | AddON | MSP | — | — | 25.8 | 55.8 | 76.9 | 68.5 | 84.5 | 65.2 |
| | | OpenMax | 3.5 | 5.9 | 33.6 | 59.1 | 76.4 | 70.5 | 87.2 | 78.8 |
| | | MaxLogits | 2.3 | 4.3 | 33.0 | 62.5 | 76.9 | 71.5 | 89.2 | 79.7 |
| | | PostMax | 8.3 | 11.0 | 34.5 | 63.4 | 76.9 | 71.8 | 89.4 | 79.2 |
| | | GHOST | 10.1 | 10.6 | 40.1 | 64.4 | 76.9 | 72.2 | 90.3 | 79.0 |
| | OE | MSP | — | 5.0 | 20.9 | 54.8 | 74.3 | 66.9 | 84.5 | 71.6 |
| | | OpenMax | — | 3.3 | 21.5 | 55.1 | 74.5 | 67.1 | 84.6 | 72.2 |
| | | MaxLogits | 2.5 | 2.7 | 23.1 | 55.7 | 74.3 | 66.9 | 85.1 | 71.0 |
| | | PostMax | 2.3 | 8.3 | 24.5 | 57.8 | 74.3 | 67.9 | 85.9 | 73.3 |
| | | GHOST | 4.7 | 10.3 | 27.3 | 57.9 | 74.3 | 67.6 | 85.9 | 71.5 |
| | OS | MSP | 3.1 | 4.4 | 25.1 | 56.5 | 74.9 | 67.8 | 85.1 | 71.2 |
| | | OpenMax | 4.2 | 4.7 | 25.7 | 57.1 | 74.9 | 67.9 | 85.3 | 71.1 |
| | | MaxLogits | 6.3 | 6.5 | 26.1 | 58.1 | 74.9 | 68.0 | 85.9 | 71.2 |
| | | PostMax | 5.8 | 9.3 | 27.9 | 58.6 | 74.9 | 68.4 | 85.8 | 73.9 |
| | | GHOST | 7.5 | 10.0 | 32.3 | 58.5 | 74.9 | 68.1 | 86.0 | 70.2 |
| $P_3$ | CE | MSP | — | — | 21.4 | 64.9 | 85.7 | 78.1 | 86.4 | 78.2 |
| | | OpenMax | — | — | 20.9 | 64.6 | 85.3 | 77.9 | 86.4 | 78.0 |
| | | MaxLogits | 1.2 | 3.2 | 14.6 | 62.6 | 85.7 | 77.9 | 87.7 | 78.1 |
| | | PostMax | 0.1 | 1.9 | 17.7 | 66.3 | 85.7 | 78.9 | 88.7 | 79.1 |
| | | GHOST | 0.4 | 2.3 | 20.1 | 65.9 | 85.7 | 78.8 | 88.6 | 78.7 |
| | ARPL | MSP | — | 6.8 | 21.5 | 64.5 | 86.5 | 78.4 | 86.0 | 78.0 |
| | | OpenMax | — | 6.8 | 21.7 | 64.4 | 86.3 | 78.3 | 86.0 | 78.0 |
| | | MaxLogits | 0.5 | 2.5 | 15.8 | 62.0 | 86.5 | 78.4 | 87.6 | 77.9 |
| | | PostMax | 1.4 | 4.8 | 17.4 | 65.4 | 86.5 | 79.3 | 88.6 | 79.0 |
| | | GHOST | 2.3 | 3.4 | 19.5 | 65.0 | 86.5 | 79.2 | 88.5 | 78.6 |
| | AddON | MSP | — | 3.7 | 24.5 | 65.6 | 85.6 | 77.8 | 86.5 | 70.7 |
| | | OpenMax | — | — | 22.5 | 65.6 | 85.6 | 78.5 | 87.0 | 77.1 |
| | | MaxLogits | 1.0 | 3.2 | 19.1 | 66.0 | 85.6 | 78.7 | 88.5 | 77.5 |
| | | PostMax | 2.1 | 4.2 | 23.5 | 69.1 | 85.6 | 79.7 | 89.5 | 79.4 |
| | | GHOST | 0.5 | 4.7 | 25.4 | 68.8 | 85.6 | 79.6 | 89.4 | 78.5 |
| | OE | MSP | — | — | 21.8 | 66.9 | 84.6 | 77.7 | 87.5 | 72.6 |
| | | OpenMax | — | — | 21.7 | 66.7 | 84.7 | 77.8 | 87.5 | 72.8 |
| | | MaxLogits | 1.1 | 2.7 | 17.7 | 65.0 | 84.6 | 77.1 | 87.2 | 72.7 |
| | | PostMax | 0.3 | 1.5 | 12.7 | 61.5 | 84.6 | 77.0 | 86.8 | 75.5 |
| | | GHOST | 0.1 | 2.6 | 22.6 | 67.3 | 84.6 | 77.8 | 87.8 | 73.8 |
| | OS | MSP | — | — | 26.6 | 67.4 | 84.9 | 78.1 | 87.8 | 75.2 |
| | | OpenMax | — | — | 26.1 | 67.2 | 84.8 | 78.0 | 87.8 | 75.4 |
| | | MaxLogits | 0.8 | 2.7 | 20.1 | 66.9 | 84.9 | 77.6 | 87.6 | 74.7 |
| | | PostMax | — | 1.4 | 11.4 | 60.6 | 84.9 | 76.9 | 86.3 | 71.4 |
| | | GHOST | 1.1 | 4.8 | 25.2 | 68.5 | 84.9 | 78.3 | 88.4 | 74.9 |

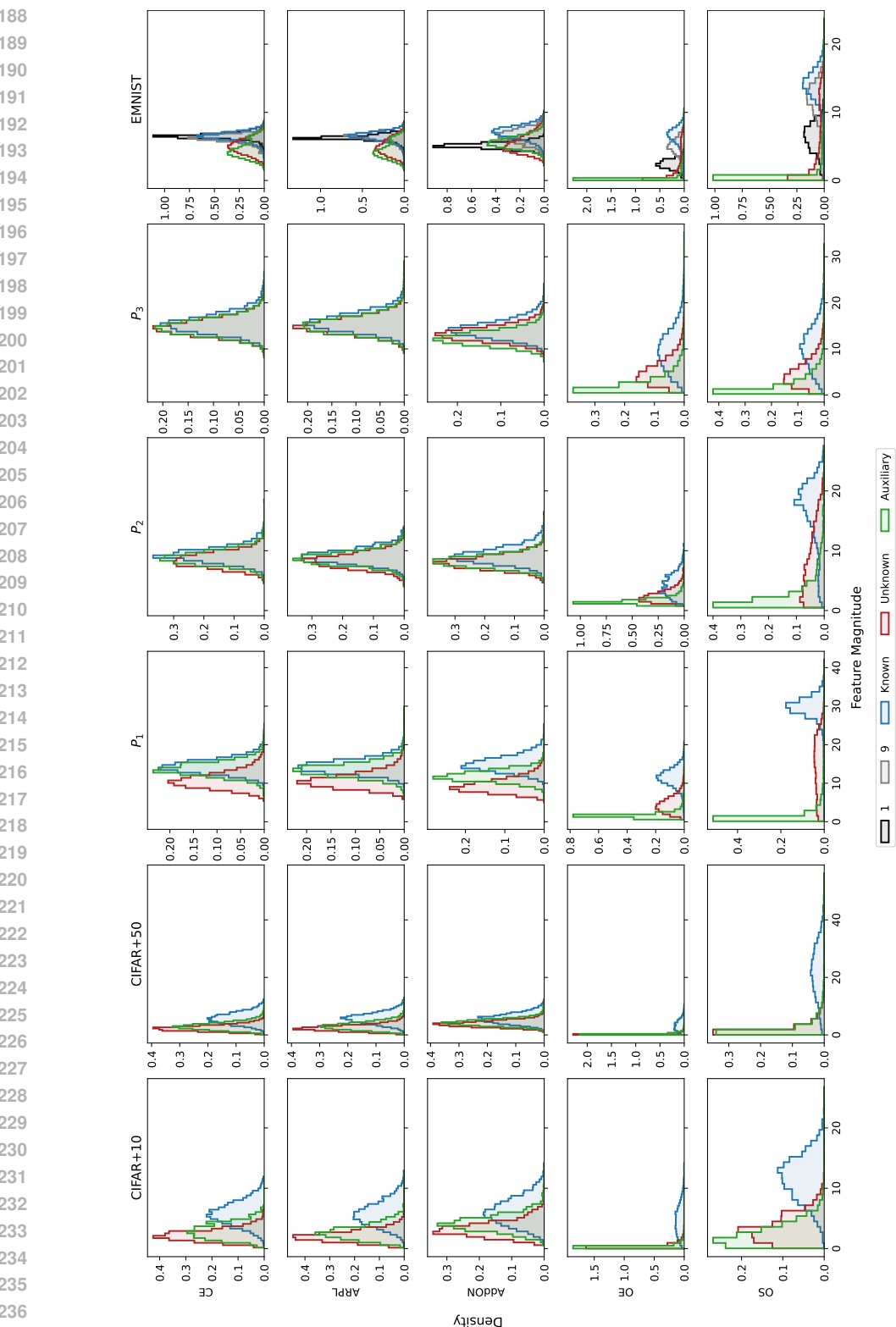

Figure 10: Feature magnitude distributions of known, auxiliary, and unknown classes on all protocols and for all RL methods, complete version of Figure 4. OE and OS experience feature magnitude collapse on $P_3$ and EMNIST, pulling their feature magnitudes towards zero. For EMNIST, we show distributions for known classes 1 and 9 (black and grey) that are highly similar to auxiliary classes, and other known classes (blue) separately.

Table 5: EMNIST ranges of class-wise CCR at the operational threshold, $\text{CCR}_c(\theta^\star)$, for classes 1, 9, and rest (all other known classes) per RL method.

| Model | Class 1 | | Class 9 | | Knowns Except 1 and 9 | |
|-------|------|------|------|------|------|------|
| | min | max | min | max | min | max |
| CE | 94.1 | 97.6 | 89.5 | 91.9 | 86.7 | 97.9 |
| ARPL | 86.7 | 96.9 | 89.5 | 92.2 | 83.8 | 98.0 |
| AddON | 67.0 | 94.8 | 92.4 | 95.6 | 95.2 | 99.5 |
| OE | 65.7 | 86.6 | 94.2 | 97.2 | 96.7 | 99.8 |
| OS | 58.1 | 88.7 | 93.6 | 95.3 | 97.7 | 99.6 |

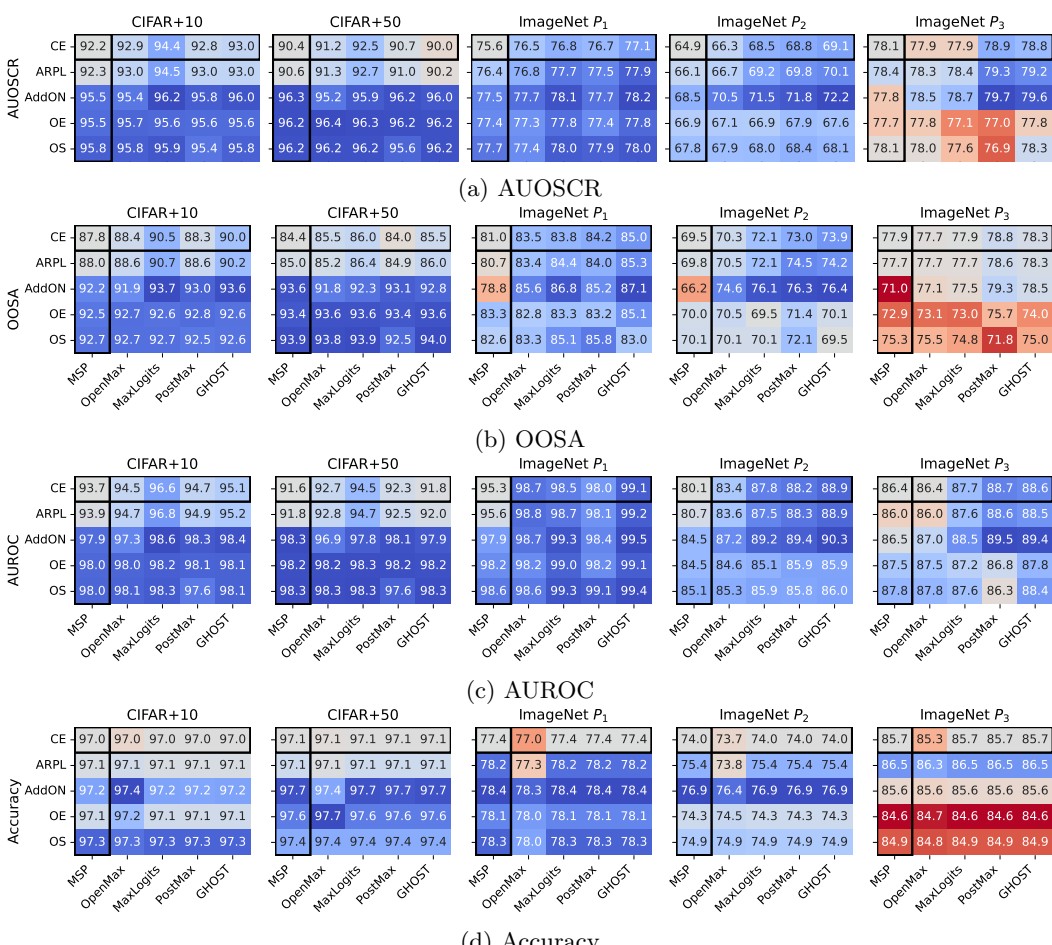

Figure 11: Theas heatmaps show the absolute values for evaluations of (a) AUOSCR, (b) OOSA, (c) AUROC, and (d) Accuracy. Each heatmap is normalized independently and centered around CE+MSP, where blue shows an increase and red a decrease. Baseline RL and PP methods are surrounded by a black border. Results for CIFAR+N are averaged over 5 trials.

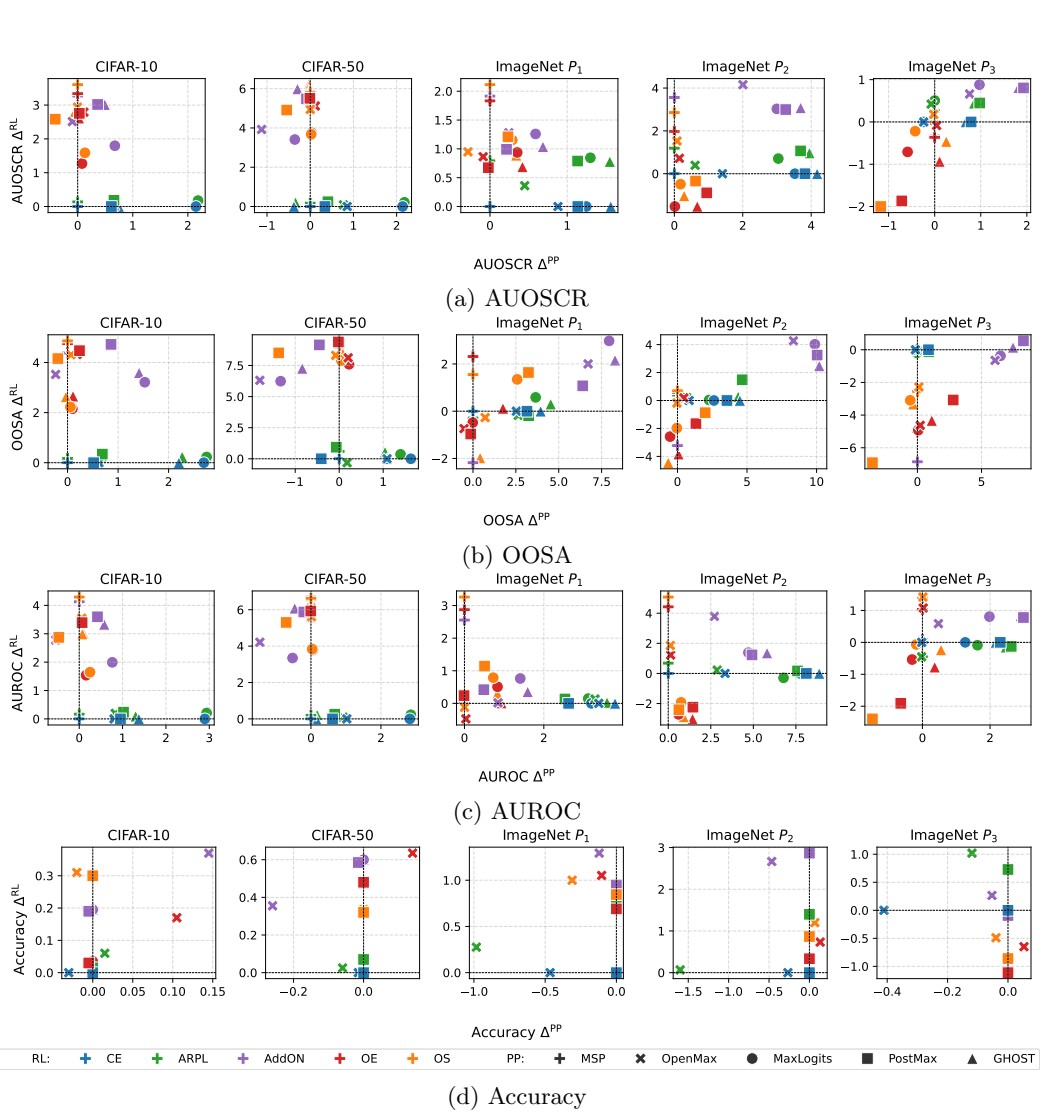

Figure 12: This figure shows the interaction effects of RL and PP components as correlation between RL performance contribution $\Delta^{\mathrm{RL}}$ and PP contribution $\Delta^{\mathrm{PP}}$ in terms of (a) AUOSCR, (b) OOSA, (c) AUROC, and (d) Accuracy. Results for CIFAR+N are averaged over 5 trials.

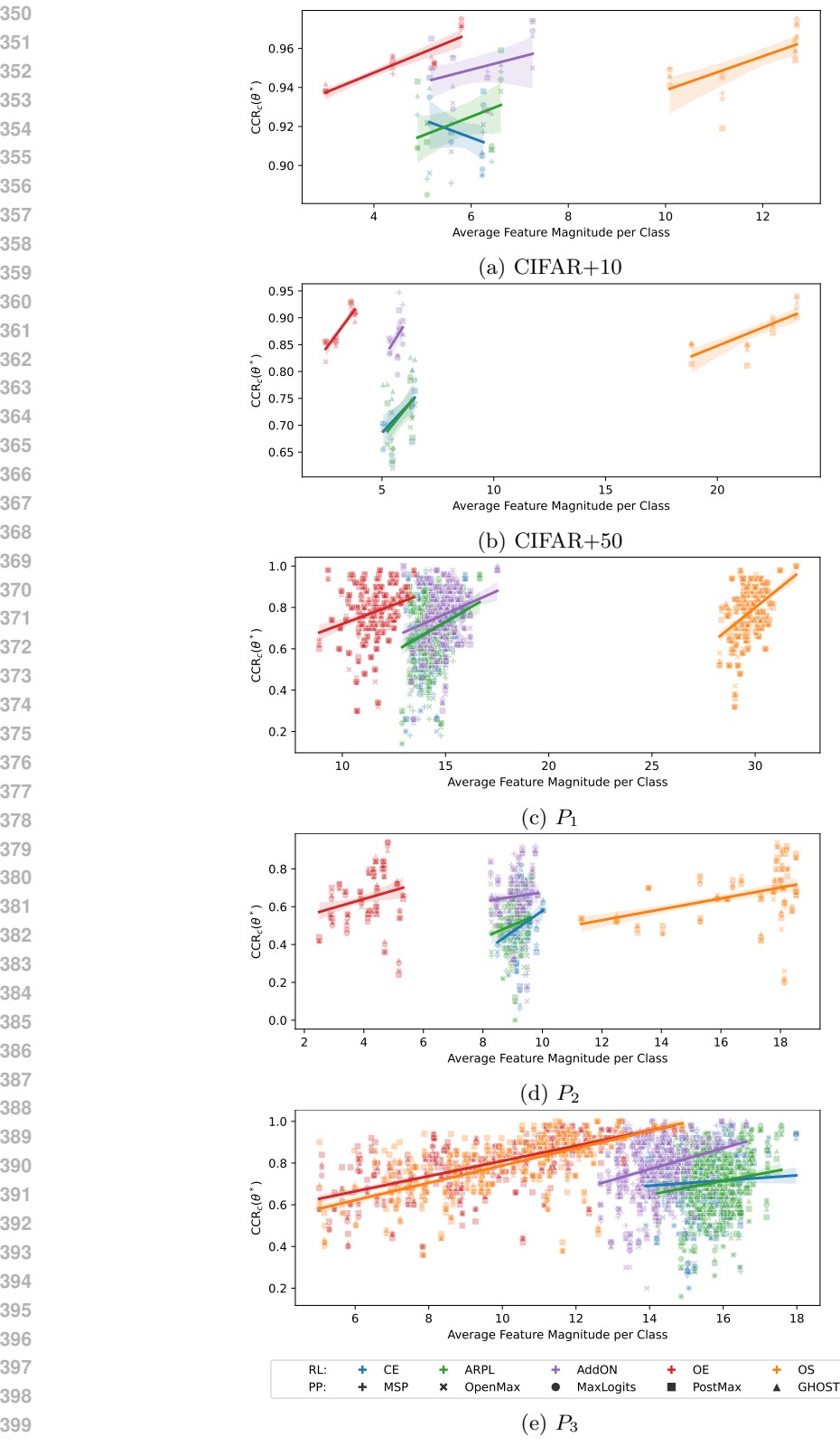

(a) CIFAR+10

(b) CIFAR+50

(c) $P_1$

(d) $P_2$

(e) $P_3$

Figure 13: Linear regressions of class-wise CCR at the operational threshold, $\mathrm{CCR}_c(\theta^*)$, against class-wise average feature magnitude for known classes. Regression is performed for each RL method independently and over all postprocessors. For CIFAR+N results are reported for the first trial only.

