# OpenReview forum: "Open-Set Recognition Interaction Effects: Modular Gains and Where to Find Them"
_ICLR.cc/2026/Conference — Submitted to ICLR 2026_

### Official Review · Reviewer_rzub · 2025-10-23

**Soundness:** 3
**Presentation:** 3
**Contribution:** 3
**Rating:** 6
**Confidence:** 3

**Summary:**

The paper studies interaction effects between representation learning (RL) and post-processing (PP) in open-set recognition (OSR) using a modular two-stage framework (RL+PP). Across small-scale (CIFAR+N) and large-scale (ImageNet P1–P3) protocols, it shows that auxiliary-data RL methods, which manipulate feature magnitudes (e.g., OE, ObjectoSphere), can degrade at scale due to a newly identified failure mode, magnitude collapse, feature norms of some known classes shrink toward the origin when auxiliary and known classes are semantically similar, yielding imbalanced class-wise CCR and poorer OSR despite gains on small datasets. Conversely, non-magnitude-manipulating RL (notably AddON, i.e., a K+1 background class) synergizes with magnitude-aware PP (e.g., PostMax, GHOST) to produce additive gains.

The key contributions are: (i) the first systematic analysis of RL-PP modular interactions, (ii) discovery and analysis of magnitude collapse, and (iii) practical guidance showing AddON + MA-PP as a robust recipe and that small-scale auxiliary-based evaluations are not predictive of large-scale performance.

**Strengths:**

Regarding originality, this is the first systematic analysis of interaction effects between RL and PP for OSR, framing OSR as a modular two-stage pipeline and introducing the magnitude-collapse failure mode, which explains why magnitude-manipulating RL degrades at scale when auxiliary and known classes are semantically similar.

In terms of quality, the study design is meticulous: five RL × five PP methods are combined, trained (mostly) from scratch to prevent leakage, and evaluated across both small-scale CIFAR+N and large-scale ImageNet P1–P3 protocols, using appropriate OSR metrics and a clear decomposition of RL versus PP gains, enabling fair attribution of effects.

On clarity, the paper clearly formalizes the RL+PP decomposition, defines decision rules, and provides intuitive visual and quantitative evidence (feature-magnitude distributions, regression linking class-wise CCR to feature norms, and heatmaps of AUOSCR/OOSA) that make the interaction story understandable. On significance, the results provide actionable guidance: non-MM RL, such as AddON combined with magnitude-aware PP (PostMax, GHOST), gives additive gains, whereas MM RL paired with MA PP should be avoided at high similarity; moreover, small-scale auxiliary-based wins do not predict large-scale behavior, which has immediate implications for benchmarking and deployment practices in OSR.

**Weaknesses:**

Although the authors provide a thorough modular evaluation, all large-scale experiments utilize a single ResNet backbone, leaving uncertainty about whether the identified interaction effects and magnitudes persist across other architectures, such as transformers, ConvNeXt, or contrastive self-supervised encoders. Incorporating these architectures would clarify if the observed norm-related behavior stems from the backbone’s feature geometry or from the learning principle itself.

 Moreover, statistical rigor is lacking: most large-scale results appear single-seeded without confidence intervals or significance tests, and several reported gains fall within plausible noise margins. Multi-seed averages, confidence intervals, and effect-size reporting would strengthen reliability.

The analysis of magnitude collapse, while intuitively presented, remains purely correlational; providing a geometric or probabilistic explanation of how auxiliary similarity drives norm shrinkage would deepen insight. Additionally, the work could explore hyperparameter sensitivity and mitigation strategies for the collapse phenomenon by systematically varying λ and ξ in OE/OS and visualizing stability regions. Baseline fairness could be improved through compute-normalized comparisons and consistent tuning across RL/PP methods.

Finally, the practical impact would benefit from an automatic diagnostic that detects early signs of magnitude collapse or misaligned RL–PP pairings during training.

**Questions:**

1. Do the RL–PP interaction patterns and magnitude behavior persist with modern backbones beyond ResNet (e.g., ViT/DeiT, ConvNeXt, Swin) and with contrastive self-supervised encoders (e.g., MoCo-v3, DINOv2)?

2. Are the reported improvements stable across random seeds and training noise, and which results remain significant after controlling for multiple comparisons across the RL×PP grid?

3. What concrete mechanism links auxiliary-known semantic similarity to feature-norm shrinkage and class-wise CCR imbalance, beyond observed correlations?

4. Can collapse be prevented or reduced by tuning OE/OS hyperparameters (λ, ξ) or by simple regularizers (norm floors, margin constraints, temperature scaling)?

5. Are baseline methods trained and tuned under comparable compute, data budgets, and augmentation/search spaces, and could budget asymmetries explain small gains?

6.  Can practitioners detect early during training when a given RL–PP pairing is at risk of magnitude collapse or harmful interaction?

---

> ### Author Response · Authors · 2025-11-25
>
> We would like to thank the reviewer for their constructive feedback and positive comments on our work. We have carefully addressed all concerns and questions raised. Please find a revised version of the paper uploaded, with changes marked in red for easier identification.
>
> - **Weakness 1 and Question 1**: We agree that this is an important point. We ran experiments with Swin-B on $P_2$ (see Fig. 7 and l.511 ff). Due to computational constraints we will only be able to run these experiments on $P_2$ which balances difficulty and computational cost.
>     1. Results show similar behavior as with ResNet-50, with feature magnitude collapse for OE and OS as well as strong performance of AddON combined with MA PP methods. This indicates that the observed interaction effects and magnitude collapse are not architecture-specific. Notably, ARPL training was unstable and did not converge with our a priori chosen seed.
>     2. Self-Supervised Methods: We believe training methods like DINOv2 or MoCo-v3 is likely unfeasible on our relatively small ImageNet subset protocols, as they typically require much larger data volumes and batch sizes.
> - **Weakness 2 and Question 2:** We agree that statistical significance is important to be tested for. That is why we did it for small scale experiments, but unfortunately we do not have the computational budget. We would like to point out that -- for the same reasons -- other large-scale studies that report on ImageNet-scale training often only report one result and do not evaluate statistical significance. Running 5 repeated trials for 9 RL methods (incl. hyperparameter optimization for OS) would require an estimated 30 to 37 GPU weeks. We encourage follow-up studies with higher computational budget and will open-source our code where running repeated trials can easily be done (see our small-scale experiments).
>
>     | Protocol | without aux data | with aux data |
>     | --- | --- | --- |
>     | $P_1$ | ~ 36 h | 55-62 h |
>     | $P_2$ | 5-9 h | 10-15 h (up to 21 h) |
>     | $P_3$ | ~ 48 h | 53-72 h (up to 84 h) |
>
> - **Weakness 3 and Question 3**: For the probabilistic argumentation, why adding auxiliary samples lead to smaller feature magnitudes, we refer to Dhamija et al. (2018). To visualize the effects of auxiliary data on the learning of known classes, we followed their setup and trained a 2D bottleneck network on EMNIST (Fig. 9), where we can clearly see that known classes that have high similarity to auxiliary data reduce feature magnitudes. However, this 2D bottleneck representation has limited transferability to higher dimensional features, and we need to postpone this intense research on finding reasons for magnitude collapse with less-overlapping known/auxiliary data to future work.
> - **Question 4:** The primary goal of this paper is to raise awareness of RL-PP interaction effects, leaving specific mitigation strategies (like the ones you proposed) for future research.
>     1. OS Sweep: Our hyperparameter sweep for OS (Section A.4) covered 9 combinations of $\xi$ and $\lambda_\mathrm{OS}$​. We found no evidence that careful selection prevents magnitude collapse.
>     2. Trend: The only noticeable trend is a preference for smaller $\lambda_\mathrm{OS}$​ values, which is intuitive as this reduces the influence of the term encouraging low feature magnitudes for auxiliary samples. In the limit, $\lambda_\mathrm{OS}\to 0$ recovers EOS training, which is highly related to OE and still exhibits collapse. We have not analyzed different values of $\lambda_\mathrm{OE}$, but we expect that smaller values would reduce magnitude collapse as we recover CE loss in the limit $\lambda_\mathrm{OE}\to 0$.
> - **Question 5:** Comparable methods (w.r.t. whether they use auxiliary data or not) are trained and evaluated with exactly the same training budget. Compute cost differences are negligible between loss functions (see response to question 2). The only noticeable difference is that the inclusion of auxiliary samples adds a bit of training time due to the processing of additional samples. We note that our training parameters were tuned on CE (for each protocol) and we found almost identical results. All other training parameters are held constant between all runs unless stated otherwise. Parameters for RL and PP methods are optimized individually on the respective validation sets of the protocols, as reported in Table 2 shown in the appendix.
> - **Weakness 5 and Question 6**: Practitioners can visualize feature magnitudes of and auxiliary and known samples during training (for the known samples even class-wise) and see whether known class magnitudes collapse. However, a full RL-PP interaction can only be tested by running a full-blown experiment, which might be too expensive as a validation procedure.
>
> References:
> - Akshay Raj Dhamija, Manuel Günther, and Terrance E. Boult. Reducing network agnostophobia. In Advances in Neural Information Processing Systems (NeurIPS), 2018.

---

### Official Review · Reviewer_bxry · 2025-10-30

**Soundness:** 2
**Presentation:** 2
**Contribution:** 2
**Rating:** 4
**Confidence:** 3

**Summary:**

The authors propose analyzing the interactions between representation
learning (RL) and post processing (PP, post-hoc methods to add
open-set capabilities) in Open Set Recognition (OSR) in a modular
2-component structure.  Particularly they analyze (feature) magnitude
manipulation (MM) in RL and magnitude-aware (MA) methods in PP.  Some
RL methods use auxiliary data/classes to representation samples of
unknown classes.

For the analysis, they use 5 existing RL (3 used auxiliary data) and 5
existing PP methods (3 are MA) over 2 datasets.  With large-scale
data, they find using auxiliary data does not improve performance.
However, with small-scale data, using auxiliary data generally improves
performance.

To understand why MM methods with auxiliary data degrade in
performance, they analyze magnitude vs performance and find positive
correlation.  Also, increasing similarity (via the P1 to P3 protocols)
between known and auxiliary samples, MM methods learn stronger
relationships.  They find that high similarity between auxiliary and
known classes can degrade the performance of MM methods.  They call
the phenomenon magnitude collapse.  To reduce magnitude collapse, they
find Additional Output Node (AddON) for the auxiliary data is
beneficial.

Without using auxiliary data, they find that RL and PP are independent
and hence any methods from RL and PP can be paired without the
contributions from one being degraded by another.

**Strengths:**

1.  Investigating the interactions between representation learning
(RL) and post processing (PP) in Open Set Recognition (OSR) in a
modular 2-component structure is interesting.

2.  The analysis indicates that with auxiliary data, similarity between
auxiliary and known classes can degrade Magnitude Manipulation (MM)
methods.  Using AddON with MM can reduce the issue.

3.  Also, the analysis indicates that without auxiliary data, RP and PP
are independent and can be paired without interference.

**Weaknesses:**

1.  The auxiliary data are intended to represent the unknown classes,
so high similarity to known classes is generally not desirable.
Consequently, the findings are not surprising.

2.  While MM methods degrade when auxiliary data are similar to known
classes, how AddON can reduce the issue is not clear.

3.  Existing methods are analyzed, but new methods are not introduced.

**Questions:**

1.  Why does AddON reduce magnitude collapse?

---

> ### Author Response · Authors · 2025-11-25
>
> We would like to thank the reviewer for their constructive feedback and positive comments on our work. We have carefully addressed all concerns and questions raised. Please find a revised version of the paper uploaded, with changes marked in red for easier identification.
>
> - **Weakness 1**: We agree the result is intuitive, but prior work often attributes performance degradation only to auxiliary/unknown class correlation (Wang et al., 2025). We show this isn't the only factor (l. 366-368).
>     1. We believe that we provide a more actionable approach to mitigating the performance degradation than focusing on high correlation between auxiliary and unknown classes. The latter are, by definition, unknown during training and therefore not something a practitioner can control for. The similarity between auxiliary training data and known classes however is testable. Were we to know unknown classes during training, then, of course, we would want to train directly on them, but this then ceases to be an OSR problem.
>     2. Tight Decision Boundaries: Some OSR literature aims to learn tight decision boundaries around known classes using auxiliary samples (Chen et al., 2021; Dhamija et al., 2018; Neal et al., 2018). We show that similarity can be detrimental, indicating that tight boundary learning requires caution.
>     3. As for our EMNIST experiments, we not only agree that the results are unsurprising, but also intended to be so. The EMNIST experiments serve as a controlled proof-of-concept to illustrate the interaction effects in a simplified setting where we can control and verify the closeness of known and auxiliary classes more precisely than in large-scale natural image protocols.
> - **Weakness 2 and Question 1**: Thank you for bringing to our attention out that we never explicitly stated how AddON avoids feature magnitude collapse. This is indeed crucial for understanding its behavior and in particular also understanding why AddON combines well with magnitude-aware postprocessors. We have expanded on this in the revised version theoretically and also provided the complete feature magnitude plot (expanded version of Fig. 4) (Section A.2; Fig. 10) that further illustrates this because we omitted AddON from Figure 4 due to space constraints.
>     1. In short, AddON prevents magnitude collapse because of the cross-entropy training objective as it must produce large probabilities for the unknown class:
>         1. CE loss (also employed by AddON) encourages probabilities close to 1 during training
>         2. This requires that $y_c$ can be reasonably close to 1. To insure this for arbitrary precision we want for $\epsilon>0$ that the following holds: $y_c > 1-\epsilon$
>         3. This holds if and only if $z_c - \max_{c^\prime\neq c}\{y_{c^\prime}\} > \log(\beta)+\log(\gamma)$, where
>             - $z_c = \max_{c\in\{1,\ldots,K+1\}}z_{c^\prime}$
>             - $\beta = \frac{1-\epsilon}{\epsilon}$
>             - $\gamma = \sum_{c^\prime\neq c}e^{z_{c^\prime}}$
>         4. We can provide an upper bound on $\log(\gamma)\leq\max_{c^\prime\neq c}z_{c^\prime} + \log(K-1) =: l$ so that we have a sufficient condition for $y_c > 1-\epsilon$, namely, $z_c - \max_{c^\prime\neq c}\{y_{c^\prime}\} > l \Longrightarrow y_c > 1-\epsilon$, where
>         5. Given a trained classifier (linear layer) $W$ with row vector $W_c$, we arrive at the sufficient condition that ensures large enough probabilities and provides an intuition on why features must exhibit a minimal magnitude (see Equation 4 and A.2).
>     2. As such, AddON does not collapse unknowns to the origin but learns a distribution of feature magnitudes for unknowns that is similar to CE (see also our response to YC14).
> - **Weakness 3:** We did not intend on introducing a new method as the focus of this study was on analyzing interaction effects and magnitude collapse. Since AddON already provides a simple and effective baseline that avoids magnitude collapse while showing strong performance at scale and lends itself well to two-stage open-set recognition, we consider proposing a new method out of scope for this work.
>
> References:
> - Guangyao Chen, Peixi Peng, Xiangqian Wang, and Yonghong Tian. Adversarial reciprocal points learning for open set recognition. Transactions on Pattern Analysis and Machine Intelligence (TPAMI), 44(11), 2021
> - Akshay Raj Dhamija, Manuel Günther, and Terrance E. Boult. Reducing network agnostophobia. In Advances in Neural Information Processing Systems (NeurIPS), 2018.
> - Lawrence Neal, Matthew Olson, Xiaoli Fern, Weng-Keen Wong, and Fuxin Li. Open set learning with counterfactual images. In European Conference on Computer Vision (ECCV), 2018.
> - Hongjun Wang, Sagar Vaze, and Kai Han. Dissecting out-of-distribution detection and open-set recognition: A critical analysis of methods and benchmarks. International Journal of Computer Vision (IJCV), 133(3), 2025.

---

### Official Review · Reviewer_zUuP · 2025-10-31

**Soundness:** 2
**Presentation:** 2
**Contribution:** 2
**Rating:** 4
**Confidence:** 2

**Summary:**

This paper presents the first systematic study of the interaction effects between Representation Learning (RL) and Post-Processing (PP) methods in Open-Set Recognition (OSR). The authors introduce a modular, two-stage framework to analyze these combinations, identifying a key failure mode termed "magnitude collapse" that affects certain RL methods at large scale. They propose a simple yet effective baseline (AddON) to mitigate this issue and provide actionable guidelines for combining RL and PP methods.

**Strengths:**

1. The core contribution—systematically studying the interaction between RL and PP—is highly novel and impactful for the OSR field.
2. The experimental setup is rigorous and thorough.
3. The (re-)introduction and thorough evaluation of AddON as a powerful and simple baseline is a significant contribution.

**Weaknesses:**

1. While the paper's title and thesis revolve around "modular gains," the quantitative evidence for the practical significance of these gains is somewhat lacking.
2. While the paper compares to canonical RL/PP methods (e.g., OE, OpenMax), it omits recent state-of-the-art OSR approaches that may interact differently with PP methods.

**Questions:**

See the weakness.

---

> ### Author Response · Authors · 2025-11-25
>
> We would like to thank the reviewer for their constructive feedback and positive comments on our work. We have carefully addressed all concerns and questions raised. Please find a revised version of the paper uploaded, with changes marked in red for easier identification.
>
> - **Weakness 1:**
>     1. Regarding the focus on "modular gains": Thank you for bringing this to our attention. Indeed, our work focuses on exploiting synergies between RL and PP methods, and crucially on *avoiding negative synergies* (magnitude collapse) for which our chosen title could be misinterpreted. We propose to change our title to better reflect this focus: "Open-Set Recognition Interaction Effects: Modular Gains and Where *(not)* to Find Them". We believe that this more adequately reflects the content of our work with respect to both potential gains (see also our next point) and potential for performance degradation.
>     2. Practical Significance: We understand this to ask whether the magnitude of improvements is practically relevant or marginal.
>         1. Generally speaking, practical relevance is application-dependent and, therefore, impossible to evaluate in general as different applications have different requirements (e.g. maximal false positive rates, minimum closed-set accuracy, etc.).
>         2. We evaluate improvements always in the context of the individual component contributions ($\Delta^\mathrm{RL}$ and $\Delta^\mathrm{PP}$), as protocol difficulty varies greatly.
>             - On $P_2$, the gains are substantial: ARPL+MSP and AddON+MSP achieve gains of
>                 - $\Delta^{RL}_{ARPL} (MSP)= 1.2$ and
>                 - $\Delta^{RL}_{AddON} (MSP)= 3.6$ percentage points in AUOSCR over CE+MSP respectively.
>                 - PostMax achieves $\Delta^{PP}_{PostMax} (CE)= 3.9$ over CE+MSP. The combinations ARPL+PostMax and AddON+PostMax achieve combined gains of 4.9 and 6.9 over CE+MSP. While this shows diminishing returns compared to independent contributions, the additional performance gain is still not negligible.
>             - On the hard protocol $P_3$, the gains are smaller across the board, but this is consistent with the small, or even negative, component contributions (e.g. in the range of -0.4 to 0.3 for RL+MSP methods and -0.2 t0 0.8 for CE+PP methods).
> - **Weakness 2:** We very much agree that the selection of methods is crucial to the validity and relevance of our findings. As such we have aimed to carefully select a representative set of both RL and PP methods covering a broad spectrum of approaches while simultaneously including similar methods to ensure that findings are not method-specific while staying within a reasonable computational budget as our experiments are already computationally very expensive.
>     1. PP Selection: We believe that our selection of PP methods covers a representative range of canonical (OpenMax 2016, MSP 2017, MLS 2022) and current state-of-the-art methods (PostMax 2024 and GHOST 2025). Methods cover various combinations of trainable and non-trainable methods, magnitude-aware and -unaware methods, as well as methods that rely on different inputs (see Table 1).
>     2. RL Selection: The selection was constrained by three requirements essential for our modular analysis:
>         1. Must be classification-based (discriminative) and produce logits.
>         2. Must be trainable from scratch on our protocols (to prevent unknown class information leakage).
>         3. Many must utilize natural auxiliary samples as this is the core focus of our study.
>         These constraints exclude many modern approaches (e.g., generative models, self-supervised methods like DINOv2/MoCo-v3). Like other recent works (Wang et al., 2025), we chose OE (2019) and ARPL (2021) as strong, representative classification-based RL methods that satisfy our constraints.
>     3. Clarity: While we have touched upon the model selection discussion in the related works section, we agree that our reasoning was not clear and explicit enough in our initial version. As such we have added a section in the appendix (A.1.1 and A.1.2) to clarify our selection process and reasoning. In case we are missing a relevant method that satisfies our constraints we would be happy to include it in future work.
>
> References:
> - Guangyao Chen, Peixi Peng, Xiangqian Wang, and Yonghong Tian. Adversarial reciprocal points learning for open set recognition. Transactions on Pattern Analysis and Machine Intelligence (TPAMI), 44(11), 2021
> - Dan Hendrycks, Mantas Mazeika, and Thomas Dietterich. Deep anomaly detection with outlier exposure. In International Conference on Learning Representations (ICLR), 2019.
> - Hongjun Wang, Sagar Vaze, and Kai Han. Dissecting out-of-distribution detection and open-set recognition: A critical analysis of methods and benchmarks. International Journal of Computer Vision (IJCV), 133(3), 2025.

---

### Official Review · Reviewer_YC14 · 2025-11-02

**Soundness:** 2
**Presentation:** 3
**Contribution:** 2
**Rating:** 4
**Confidence:** 4

**Summary:**

This paper systematically analyzes Open-Set Recognition (OSR) as a modular, two-stage framework combining Representation Learning (RL) and Postprocessing (PP). Its central contribution is the discovery of "magnitude collapse," a failure mode where popular magnitude-manipulating (MM) methods like Outlier Exposure fail at scale. The authors show this occurs when high similarity between known and auxiliary data causes the model to irreversibly destroy feature magnitude information. They contrast this with the simple, non-MM AddON ($K+1$ classifier), which remains robust. The paper's method is to study the "interaction effects" between different RL and PP components, concluding that small-scale benchmarks are misleading and that robust performance comes from the correct combination of methods (e.g., AddON + PostMax), not a single "best" component.

**Strengths:**

- Key Insight on Feature Magnitude: The paper's primary strength is its insightful diagnosis of the "magnitude collapse" failure mode. This provides a clear, testable explanation for why popular methods that excel on small-scale benchmarks (like OE) fail on large-scale ones, linking the failure to high semantic similarity between known and auxiliary data.

 - Marginal SOTA Improvement: While the paper successfully identifies a robust combination (AddON + PostMax), the resulting performance gains over other strong, existing combinations (e.g., CE + GHOST or ARPL + GHOST) are often marginal. For instance, on the $P_3$ benchmark (Figure 2, AUOSCR), the proposed AddON+PostMax (79.7) is only a minor improvement over ARPL+GHOST (79.2), suggesting the baseline was already very strong.

**Weaknesses:**

- Limited Compatibility of AddON: The AddON method's objective creates a representation that is incompatible with many existing OOD detectors. AddON trains the model to produce high-magnitude signals for unknowns (at its $K+1$ node), which directly contradicts the core assumption of many feature-norm-based detectors that expect low-magnitude signals for unknowns. This limits the "modular" combinations to only those PP methods that can be adapted to AddON's specific logic.

 - Oversimplification of the Unknown Space: The AddON method relies on a $K+1$ classifier, which fundamentally models the entire, infinitely diverse "unknown" space using a single prototype vector (the weights for that node). This is a significant oversimplification that likely only works on benchmarks where the unknown classes have limited diversity or happen to be well-represented by the specific auxiliary data used.

- Entangled Evaluation Metrics: The paper's main metrics for OSR performance (AUOSCR and OOSA) entangle unknown detection performance and closed-set classification accuracy into a single score. This can be misleading, as a method could improve in one aspect while regressing in the other. For example, the paper's results in Figure 8 show that AddON has a slightly worse closed-set accuracy than ARPL on ImageNet $P_3$, but its detection (AUROC) is stronger. The final AUOSCR score obscures this trade-off.

**Questions:**

Please refer to the above weaknesses

---

> ### Author Response · Authors · 2025-11-25
>
> We would like to thank the reviewer for their constructive feedback and positive comments on our work. We have carefully addressed all concerns and questions raised. Please find a revised version of the paper uploaded, with changes marked in red for easier identification.
>
> - **Strength 2:** We acknowledge the small marginal improvements from RL methods on $P_3$​, which is a notoriously difficult protocol (Palechor et al. (2023)). The primary value is the modular improvement from combining methods, like the total 1.6 percentage point increase in AUOSCR achieved by combining GHOST with CE and AddON. Improvements are more pronounced on other protocols.
> We've clarified the baselines in the revised version (Fig. 2 heatmaps bordered). ARPL + GHOST is not an existing baseline as we are the first to combine them. Existing baselines are CE+PP (for postprocessors) and RL+MSP (for representation learning methods).
> - **Weakness 1:** We agree and have expanded the discussion on AddON (see l.403ff and A.2).
>     1. AddON vs. MM methods: AddON is similar to CE in that it produces "sufficiently large" feature magnitudes for known, auxiliary, and unknown samples (no collapse to the origin). It surprisingly learns lower feature magnitudes for auxiliary/unknown classes compared to known classes, but maintains an overall large magnitude and similar distribution characteristics to CE (Fig. 10). This makes its representations compatible with MA PP methods.
>     2. Prior Work: We emphasize that we did not invent AddON (studied under various names (see Appendix A.1.1)), but we demonstrate its effectiveness and robustness at scale, especially in combination with MA PP methods.
> - **Weakness 2:** We agree that this is a very strong simplification, even more so as it is unclear what features are supposed to be learned from a diverse set of auxiliary samples. Yet this simplicity empirically proves surprisingly effective and robust to various datasets.
>     1. Diversity & Robustness: We can confidently say that AddON's good performance on P1​ (diverse unknowns: non-animal classes; auxiliary samples: 4-legged non-dog animals; knowns: dog classes) confirms that it is not dependent on limited diversity or auxiliary samples closely representing unknowns.
>     2. Closed-set Accuracy: This (over)simplification also does not harm closed-set accuracy compared to CE, even leading to improvements on $P_1$ and $P_2$ (See CCR@100 in Table 2 in the Supplemental material).
>     3. Future Work: We chose to stick to the basic, widely-used single-node approach for this study, as we do not claim to have invented AddON. Future work could investigate more sophisticated unknown modeling (e.g., multiple output nodes *K+U*).
>     4. High Dimensionality: In our models, the high-dimensional feature space (D=512 or D=2048) relative to the known classes (K=4 or K≤151) can still allow the unknowns to occupy a broad, representative space.
> - **Weakness 3:** ting classification and OOD detection performance. We address this by reporting and discussing closed-set accuracy and AUROC in the appendix, highlighting their contributions (cf. l.368-371 and l.398-402).
>     1. Advantages of Aggregate Metrics:
>         1. They enable model selection (e.g., hyperparameter optimization on validation set, see A.3 and A.4) and serve as a target for our regression analysis (Section 4.4).
>         2. They jointly capture classification and OOD detection performance for any given threshold. Limiting our evaluation to only closed-set accuracy and AUROC fails to capture crucial interaction effects between $k^\star$ (known class prediction) and $\gamma^\star$ (OOD score). AUOSCR/OOSA capture performance given a threshold for OOD detection.
>     2. Established metrics: Metrics like AUOSCR are established methods for OSR evaluation alongside accuracy and AUROC, which is why we include them.
>     3. **Response to Reviewer Question**: We recognize this is a widely discussed trade-off (Dhamija et al. (2018), Wang et al. (2022), etc.). We've already addressed this by reporting and discussing all four metrics. We believe the issue is likely a lack of explicit clarity. We have expanded the discussion in Section 4 on why we focus on standard OSR metrics, detailing the advantages and disadvantages of each, and further discussing all metrics in Appendix A.4.
>
> References:
> - Akshay Raj Dhamija, Manuel Günther, and Terrance E. Boult. Reducing network agnostophobia. In Advances in Neural Information Processing Systems (NeurIPS), 2018.
> - Andres Palechor, Annesha Bhoumik, and Manuel Günther. Large-scale open-set classification protocols for ImageNet. In Winter Conference on Applications of Computer Vision (WACV), 2023.
> - Zitai Wang, Qianqian Xu, Zhiyong Yang, Yuan He, Xiaochun Cao, and Qingming Huang. OpenAUC: Towards AUC-oriented open-set recognition. In Advances in Neural Information Processing Systems (NeurIPS), 2022.

---

### Meta-Review · Area_Chair_pv4u · 2026-01-06

**Summary:**

The paper presents a systematic study  of Open-Set Recognition (OSR) by decomposing it into Representation Learning (RL) and Post-Processing (PP) stages. The authors identify a failure mode called "magnitude collapse" in magnitude-manipulating (MM) methods when applied to large-scale data with high semantic similarity between known and auxiliary classes. They propose that a simple baseline, AddON, combined with magnitude-aware PP (like PostMax), is more robust than complex MM methods.

**Reviewer Concerns:**

Reviewers appreciate the systematic analysis of RL-PP interactions and the identification of "magnitude collapse" as a novel contribution. However, they criticize the lack of theoretical depth behind the failure mode, the marginal performance gains over existing strong baselines, and the reliance on a single backbone (ResNet).

**Reviewer Scores:**

This paper recieves mixed scores. after reading the rebuttals, The AC aggree with the reviewers that the method gives marginal gains: the quantitative improvement of the proposed combination (AddON + PostMax) over existing strong baselines (e.g., ARPL + GHOST) is very small (e.g., 79.7 vs 79.2 AUOSCR), and Lack of Theoretical Depth: the explanation for "magnitude collapse" is correlational rather than causal or geometric, and Limited evaluation experiments: only evaluated on a single ResNet backbone.

---

### Decision · Program_Chairs · 2026-01-26

Reject